# Your VAR Model is Secretly an Efficient and Explainable Generative Classifier

**Yi-Chung Chen, David I. Inouye, Jing Gao**
Elmore Family School of Electrical and Computer Engineering
Purdue University
{chen5262, dinouye, jinggao}@purdue.edu

## Abstract

Generative classifiers, which leverage conditional generative models for classification, have recently demonstrated desirable properties such as robustness to distribution shifts. However, recent progress in this area has been largely driven by diffusion-based models, whose substantial computational cost limits their scalability in practice. To address the efficiency concern, we investigate generative classifier built upon recent advances in visual autoregressive (VAR) modeling. Owing to their tractable likelihood, VAR-based generative classifier enable significantly more efficient inference compared to diffusion-based counterparts. Building on this foundation, we introduce the Adaptive VAR Classifier$^+$ (A-VARC$^+$), which further improves accuracy while reducing computational cost, substantially enhancing practical usability. Beyond efficiency, we also study several properties of VAR-based generative classifiers that distinguish them from conventional discriminative models. In particular, the tractable likelihood facilitates visual explainability via token-wise mutual information, and the model naturally adapts to class-incremental learning without requiring additional replay data.

## 1 Introduction

Generative models are trained to directly capture the underlying data distribution of a given dataset, which enables a wide range of applications such as image generation (Han et al., 2025), image editing (Mu et al., 2025), and data augmentation (Trabucco et al., 2023). Given this expressive capability, a natural question arises: **Can we leverage these powerful generative models for classification?** This question has motivated a line of research on the "Generative Classifier." In this paradigm, class-conditional generative models are employed to estimate the likelihood $p(x|y)$, where $y$ denotes the class label and $x$ the input data. The posterior distribution $p(y|x)$ can then be derived via Bayes' theorem, given the prior $p(y)$. This stands in contrast to conventional discriminative classifiers, which directly model the conditional probability $p(y|x)$. Although generative classifiers are less commonly adopted due to the inherent difficulty of accurately modeling $p(x|y)$—a substantially harder task than modeling $p(y|x)$—prior work has shown that they exhibit several advantageous properties distinct from discriminative classifiers.

Early work on generative classifiers (Schott et al., 2019) employed VAE (Kingma & Welling, 2013) to model the likelihood $p(x|y)$ and demonstrated that such classifiers exhibit greater robustness against adversarial attacks compared to discriminative models on the MNIST dataset (Deng, 2012), a finding further supported by Li et al. (2019). In addition, Van De Ven et al. (2021) showed that such generative classifiers achieve superior performance in class-incremental learning. Building on these successes, subsequent studies explored normalizing flows (Ardizzone et al., 2020; Mackowiak et al., 2021) and score-based models (Zimmermann et al., 2021) for estimating class-conditional likelihoods, achieving classification performance comparable to discriminative counterparts. More recent work adapted pre-trained text-to-image diffusion models as zero-shot generative classifiers by approximating likelihood through the evidence lower bound (ELBO), revealing desirable properties such as out-of-distribution robustness (Li et al., 2023), attribute binding (Clark & Jaini, 2023), shape bias, and human-like error consistency (Jaini et al., 2023). Further studies have shown that diffusion-based generative classifiers can achieve certifiable robustness (Chen et al., 2024a) and are robust to subpopulation shifts (Li et al., 2024), underscoring their promising advantages.

Despite these advances, research on generative classifiers for image classification remains relatively underexplored. The recent emphasis on diffusion-based generative classifiers introduces two limitations. The first and most critical challenge is scalability. By design, generative classifiers suffer from efficiency issues, as their computational complexity grows linearly with the number of classes, which severely limits applicability to large-scale datasets such as ImageNet with 1,000 classes. This issue is further exacerbated by the lack of tractable likelihoods in diffusion models. Specifically, the diffusion-based method adopts ELBO as an approximation of likelihood (Li et al., 2023), which involves multiple forward passes. To obtain reliable approximations for classification, diffusion-based methods typically require dozens to hundreds of function evaluations, creating a significant barrier to practical deployment. The second limitation is the narrow perspective that arises from the recent exclusive focus on diffusion-based approaches. It is unclear whether the desirable properties reported in diffusion-based studies are shared across different generative classifiers.

Recent advances in visual autoregressive (VAR) modeling (Tian et al., 2024) present a promising and efficient backbone for generative classifiers. However, a naive implementation of a VAR classifier (VARC) yields suboptimal performance. To address this, we propose the Adaptive VAR Classifier (A-VARC), a framework designed to improve accuracy while reducing computational cost. A-VARC integrates two techniques: **(i) likelihood smoothing**, which enhances accuracy by producing more robust likelihood estimates, and **(ii) partial-scale candidate pruning**, which accelerates inference by exploiting the model's multi-scale architecture for candidate pruning. Together, these components form a flexible and efficient generative classifier that outperforms the naive VARC baseline. In addition, we introduce A-VARC$^+$, an enhanced variant finetuned using the recently proposed Condition Contrastive Alignment (CCA) method (Chen et al., 2024b). With these improvements, A-VARC$^+$ achieves accuracy comparable to the DiT-based (Peebles & Xie, 2023) diffusion classifier on ImageNet-100—incurring less than a 1% drop—while requiring 89× less computational cost. This dramatically reduces the computational burden of generative classifiers and significantly improves their practical feasibility.

Beyond efficiency, we also investigate additional properties of VAR-based generative classifiers. Although we do not observe the same level of distribution-shift robustness reported for diffusion-based approaches, our analysis uncovers other distinctive advantages. In particular, the tractable likelihood enables visual explanations via token-wise mutual information, capturing the relevance of individual tokens to the target label. Moreover, unlike discriminative classifiers that typically suffer from catastrophic forgetting in class-incremental learning, VAR-based models, trained to model class-conditional likelihoods independently, naturally adapt to such tasks without requiring replay data. Collectively, these findings highlight new and complementary research directions for generative classifiers.

The main contributions of our paper are summarized as follows:

- We investigate VAR-based generative classifiers and introduce A-VARC$^+$, which further improves both performance and efficiency. Notably, A-VARC$^+$ achieves accuracy comparable to the DiT-based diffusion classifier while requiring 89× less computational cost.

- We conduct a comprehensive evaluation of generative classifiers across multiple model families and datasets under a well-controlled setup, providing a clearer understanding of their strengths and limitations.

- We show that the tractable likelihood of VAR-based generative classifiers enables visual explainability and allows the model to naturally adapt to class-incremental learning without the need for replay data.

## 2 RELATED WORK

### 2.1 GENERATIVE CLASSIFIER

The discussion of generative classifiers can be traced back to Ng & Jordan (2001), who studied Naive Bayes and showed its superior data efficiency compared to its discriminative counterpart. Subsequent research has explored generative classifiers built upon different backbone architectures. Early works (Schott et al., 2019; Li et al., 2019; Ghosh et al., 2019) employed VAEs to model the likelihood and demonstrated strong adversarial robustness. Van De Ven et al. (2021) further

showed that such classifiers achieve superior performance in class-incremental learning. Follow-up studies investigated normalizing flows. Fetaya et al. (2020) highlighted a key limitation of conditional likelihood–based generative classifiers, noting that class-conditional information may be underutilized when trained with a maximum-likelihood objective. Ardizzone et al. (2020) showed that this issue can be alleviated by introducing a reweighted discriminative term, and Mackowiak et al. (2021) demonstrated that such a design enables additional features such as explainability and out-of-distribution detection.

More recently, the rapid progress of diffusion models has motivated their adoption for generative classification. Zimmermann et al. (2021) derived class-conditional likelihoods via reverse SDE, showing improved performance on CIFAR-10 (Krizhevsky et al., 2009). Alternatively, Li et al. (2023) employed the ELBO as a proxy for likelihood estimation, demonstrating robustness to distribution shifts. Follow-up works (Clark & Jaini, 2023; Jaini et al., 2023) further highlighted intriguing properties such as human-like shape bias and error consistency with human judgments. Recent studies extended these findings by showing that ELBO-based diffusion classifiers can achieve substantial improvements in certified robustness (Chen et al., 2024a) and mitigate shortcut learning caused by spurious correlations (Li et al., 2024). While some studies have explored autoregressive generative classifiers in NLP tasks (Li et al., 2024; Kasa et al., 2025), the recent investigation of autoregressive-based generative classifiers for image classification remains limited, with Jaini et al. (2023) providing only preliminary results. In this work, we study VAR-based generative classifiers, which provide a new perspective on the development of generative classifiers.

## 2.2 Image Autoregressive Model

For image generation, autoregressive models transform the intractable problem of modeling all pixel dependencies simultaneously into a tractable sequence of prediction tasks. Larochelle & Murray (2011) demonstrated the feasibility of building neural autoregressive models for image generation. Follow-up works (Van Den Oord et al., 2016; Van den Oord et al., 2016; Salimans et al., 2017) introduced architectural improvements and performed next-pixel prediction in a raster-scan manner. The development of VQ-VAE (Van Den Oord et al., 2017) further enabled encoding images into shorter sequences of discrete tokens, greatly improving scalability. Subsequent works (Razavi et al., 2019; Ramesh et al., 2021; Esser et al., 2021; Sun et al., 2024) demonstrated the outstanding generative capability of such models. Moving beyond conventional next-token prediction, Tian et al. (2024) proposed visual autoregressive (VAR) modeling with next-scale prediction, which generates images in a multi-scale, coarse-to-fine order and achieves performance superior to earlier approaches. Building upon this advance, we show that, with its tractable likelihood and next-scale prediction, the VAR model can also serve as an efficient and explainable generative classifier.

## 3 Preliminary

### 3.1 Generative Classifier

Given an image–label pair $(x, y)$, the goal of a classifier is to model the conditional probability $p(y|x)$ for classification. Unlike discriminative classifiers, which directly learn this distribution, generative models are trained to estimate the class-conditional likelihood $p(x|y)$. Using Bayes' theorem, the parameterized posterior $p_\theta(y|x)$ can then be expressed as:

$$p_\theta(y_i \mid x) = \frac{p_\theta(x \mid y_i)p(y_i)}{\sum_j p_\theta(x \mid y_j)p(y_j)} \tag{1}$$

where $p(y_i)$ denotes the class prior. A common assumption is that the prior is uniform across all classes, in which case the prediction of a generative classifier is obtained by:

$$\arg\max_{y_i} p_\theta(x \mid y_i) \tag{2}$$

Note that performing classification using Eq. 2 requires the conditional likelihood $p_\theta(x|y_i)$ for every possible class $y_i$, and thus the computational complexity scales linearly with the number of classes.

## 3.2 Diffusion Classifier

For diffusion models, the class-conditional likelihood $p(x|y)$ is intractable. To address this, the diffusion classifier Li et al. (2023) employs the evidence lower bound (ELBO) as a surrogate objective. The classification decision can then be obtained as:

$$\arg\min_{y_i} \mathbb{E}_{t,\epsilon}\big[\|\epsilon_\theta(x_t, y_i) - \epsilon\|^2\big], \ \epsilon \sim \mathcal{N}(0, I) \tag{3}$$

where $\epsilon_\theta$ is the noise prediction model and $t$ is the timestep used for determining the noise level. The intuition is that with the correct class condition, the noise prediction error will be smaller. However, obtaining a reliable estimate typically requires dozens to hundreds of Monte Carlo samples, resulting in substantial computational overhead.

## 3.3 Image Autoregressive Model

Autoregressive models provide a principled way to represent the complex distribution of high-dimensional data by factorizing it into a product of one-dimensional conditional distributions. To model the likelihood $p(x|y)$, an image $x$ is first tokenized into a sequence of tokens $(x_1, x_2, \cdots, x_L)$. The class-conditional likelihood can then be expressed as:

$$p(x \mid y) = p(x_1, x_2, \cdots, x_L \mid y) = \prod_l^L p(x_l \mid x_1, x_2, \cdots, x_{l-1}, y). \tag{4}$$

A common tokenization strategy is to use a quantized autoencoder, such as VQ-VAE (Van Den Oord et al., 2017), which converts an image feature map $f \in \mathbb{R}^{h \times w \times C}$ into discrete tokens $q \in [V]^{h \times w}$, typically ordered in raster-scan fashion. However, Tian et al. (2024) identifies limitations of raster-scan ordering, including loss of structural information and inefficiency, and instead proposes next-scale prediction. In this approach, a feature map $f \in \mathbb{R}^{h \times w \times C}$ is quantized into $K$ multi-scale token maps $(r_1, r_2, \cdots, r_K)$, each with progressively higher resolution $h_k \times w_k$, culminating in $r_K$, which matches the original resolution $h \times w$. Specifically, for a given image $x \in \mathbb{R}^{C \times H \times W}$, the tokenization process is defined as:

$$f = \mathcal{E}(x), \quad (r_1, r_2, \cdots, r_K) = \mathcal{Q}(f), \tag{5}$$

where $\mathcal{E}(\cdot)$ denotes the encoder and $\mathcal{Q}(\cdot)$ the quantizer. The multi-scale token maps can be projected back to pixel space as a reconstructed image $\hat{x}$ through a codebook $Z$ and a decoder $\mathcal{D}(\cdot)$ as follows:

$$\hat{f} = \text{lookup}(Z, (r_1, r_2, \cdots, r_K)), \quad \hat{x} = \mathcal{D}(\hat{f}). \tag{6}$$

where $\text{lookup}(Z, \cdot)$ means taking the corresponding vector in codebook $Z$. For high-capacity VQ-VAE models, the difference between $x$ and $\hat{x}$ is generally negligible. Therefore, the VAR model is trained on the discrete token set, which is formulated as:

$$p_\theta(x \mid y) = p_\theta(r_1, r_2, \cdots, r_K \mid y) = \prod_k^K p_\theta(r_k \mid r_1, r_2, \cdots, r_{k-1}, y), \tag{7}$$

where each $r_k \in [V]^{h_k \times w_k}$ is the token map at scale $k$, containing $h_k \times w_k$ tokens.

## 4 Adaptive VAR Classifier

With the tractable likelihood defined in Eq.7, a VAR model can be directly converted into a VAR classifier (VARC) using Eq.1. Since the token maps $(r_1, r_2, \cdots, r_K)$ are readily available after tokenizing a test image with VQ-VAE, the likelihood can be estimated with a single forward pass, making VARC a more efficient classifier compared to diffusion-based methods. However, this naive adaptation yields suboptimal performance. To address this, we first introduce the Adaptive VAR Classifier (A-VARC), which integrates two key techniques: likelihood smoothing and partial-scale candidate pruning. By design, A-VARC adaptively balances accuracy and efficiency, offering significant improvements over the naive VARC baseline. We then present A-VARC$^+$, an enhanced variant that applies CCA finetuning to achieve further performance gains.

## 4.1 Likelihood Smoothing

While Eq.7 provides a formulation for likelihood estimation, we observe that it lacks smoothness and may lead to suboptimal performance. To illustrate this issue, we add a small perturbation to the feature map $f$, producing a noised feature map $f^{noise}$ and reconstructing a corresponding image $\hat{x}^{noise}$ as follows:

$$f^{noise} = f + \epsilon, \quad \hat{x}^{noise} = \mathcal{D}(\text{lookup}(Z, \mathcal{Q}(f^{noise}))), \tag{8}$$

where $\epsilon \sim \mathcal{N}(0, \sigma^2)$ is sampled from a Gaussian distribution with small variance $\sigma^2$. Although $\hat{x}$ and $\hat{x}^{noise}$ are visually almost indistinguishable (see Fig.1), their token maps $\mathcal{Q}(f)$ and $\mathcal{Q}(f^{noise})$ differ drastically, with 69% of tokens changed. This discrepancy in token maps also causes observable variations in the estimated likelihood. However, since $\hat{x}$ and $\hat{x}^{noise}$ are perceptually similar, the corresponding likelihoods should ideally differ only slightly in order to yield stable classification.

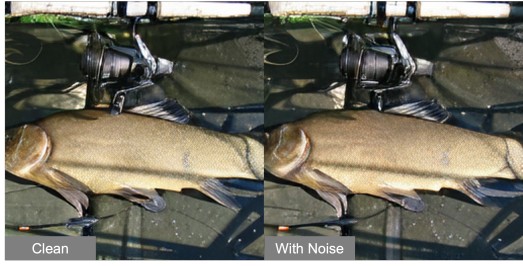

Figure 1: Visual comparison of images with and without noise perturbation.

To address this problem, we propose a smoothed class-conditional likelihood defined as:

$$\tilde{p}_{\theta,S}(x \mid y) = \sum_{i}^{S} p_\theta(\mathcal{Q}(f + \epsilon_i) \mid y), \ \epsilon_i \sim \mathcal{N}(0, \sigma^2) \tag{9}$$

where $S$ is the number of samples used for smoothing. Our empirical results demonstrate that likelihood smoothing effectively improves classification accuracy. Although this approach introduces additional computational cost, we find that using only a small value of $S$ already yields noticeable gains, and the overhead can be further mitigated through the candidate pruning strategy.

## 4.2 Partial-Scale Candidate Pruning

While the tractable likelihood of the VAR model alleviates the computational cost of likelihood estimation, the most critical efficiency challenge of generative classifiers still remains. Specifically, as shown in Eq. 2, the computation cost scales linearly with the number of classes, since classification requires evaluating the class-conditional likelihood for all possible classes. This limitation greatly restricts the applicability to datasets with large numbers of classes. To address this, prior works adopt a two-stage procedure: first, apply a quick but coarse likelihood estimation method to filter out unlikely candidates, and then use a more accurate but computationally expensive estimation on the remaining classes. For instance, Li et al. (2023) employs such a strategy by using 25 samples to approximate the ELBO in Eq. 3 for all classes, followed by 250 samples for the top-5 candidates identified in the first stage to refine the predictions.

Inspired by this idea, A-VARC adopts a similar but more aggressive candidate pruning strategy. Unlike conventional autoregressive image generation models, the VAR model employs next-scale prediction, which generates images in a coarse-to-fine multi-scale order. This design encodes global structural information at each scale with varying resolutions. Because each scale contains global information, we find that the partial information in the first few scales is often sufficient to discriminate between classes with large visual differences (e.g., distinguishing a tench from a hen). This observation motivates a more efficient pruning strategy. Specifically,

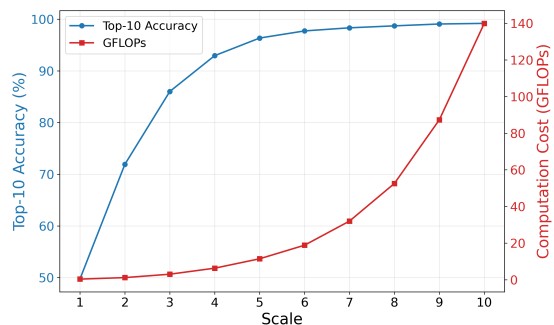

Figure 2: Top-10 accuracy and computation cost vs. number of scales.

we introduce partial-scale likelihood approximation, defined as:

$$\hat{p}_{\theta,K'}(x \mid y) = p_\theta(r_1, r_2, \cdots, r_{K'} \mid y) = \prod_k^{K'} p_\theta(r_k \mid r_1, r_2, \cdots, r_{k-1}, y) \tag{10}$$

where $K' < K$ denotes the number of scales used. Fig. 2 shows the top-10 accuracy and per-image computation cost on ImageNet-100 across different values of $K'$. The results indicate that the approximated likelihood $\hat{p}_{\theta,K'}(x|y)$ with a small $K'$ achieves comparable top-10 accuracy to the full-scale likelihood, while offering substantial efficiency improvements, making it well suited for candidate pruning. The efficiency gain stems from the reduction in token length: the full multi-scale token maps contain 680 tokens, whereas the first five scales include only 55 tokens—about 8% of the total—due to the smaller $h_k$ and $w_k$ at lower resolutions. This partial-scale pruning strategy significantly reduces the computational burden of A-VARC, allowing resources to be focused on the most likely candidates.

### 4.3 CLASS INFORMATION ENHANCEMENT VIA CCA

One possible factor contributing to the suboptimal performance of VAR-based classifiers is that class-conditional information may be underutilized when training with a maximum-likelihood objective, as discussed in (Fetaya et al., 2020). Prior work has attempted to address this limitation by incorporating an additional discriminative term into the training objective to strengthen class information (Fetaya et al., 2020; Ardizzone et al., 2020; Mackowiak et al., 2021). While this approach does enhance class-conditional information and improve classification accuracy, the improvement comes at the expense of generation ability. A similar phenomenon is observed in image generation tasks, where conditional image generation results may not strictly align with the given condition when sampling directly from the conditional distribution. A common remedy in image generation is classifier-free guidance (Ho & Salimans, 2022), which enhances conditional information by extrapolating between conditional and unconditional distributions. However, we find that this technique is ineffective for generative classifiers and, in fact, degrades performance through our empirical results and also reported by previous work Li et al. (2023). We hypothesize that this is because classifier-free guidance sharpens the density of a subset of the distribution to produce visually appealing images, but weakens the model's general likelihood estimation ability, as also argued by (Karras et al., 2024).

In this work, we want to find a general solution that enhances the class conditional information and benefits both the generation and classification tasks. Fortunately, we found that the recent advance in VAR finetuning provides an ideal solution. Chen et al. (2024b) introduced a novel finetuning objective called Condition Contrastive Alignment (CCA), which was originally proposed to enhance the class-conditional information through a finetuning technique to eliminate the necessity of classifier-free guidance. Specifically, given a pretrained conditional generative model $p_\phi$, the CCA objective encourages the target model $p_\theta$ to strengthen class-conditional information as follows:

$$\mathcal{L}_\theta^{CCA}(x, y, y^{neg}) = -\log \sigma_{sig}[\beta \log \frac{p_\theta(x \mid y)}{p_\phi(x \mid y)}] - \lambda \log \sigma_{sig}[-\beta \log \frac{p_\theta(x \mid y^{neg})}{p_\phi(x \mid y^{neg})}] \tag{11}$$

where $\sigma_{sig}(\cdot)$ denotes the sigmoid function, and $\beta$ and $\lambda$ are hyperparameters. The pretrained model $p_\phi$ remains fixed during finetuning. Intuitively, the first term encourages the model to increase the likelihood under the ground-truth label $y$, while the second term penalizes high likelihood under an incorrect label $y^{neg}$. This objective effectively reinforces class-conditional information in conditional generative models. Our empirical results demonstrate that applying CCA to A-VARC further improves classification performance by guiding the model to focus on more object-relevant regions, as illustrated in Fig. 6 in the Appendix. We denote this enhanced version as A-VARC$^+$.

## 5 COMPARATIVE ANALYSIS

In this section, we conduct detailed experiments to evaluate the performance of the proposed A-VARC$^+$ and verify if the robustness property is transferable to our VAR-based method.

**Datasets and Evaluation Metric.** We evaluate the proposed A-VARC$^+$ across a diverse set of datasets to assess its performance from multiple perspectives. For general classification ability, we

report results on ImageNet-100, a randomly sampled subset of ImageNet (Deng et al., 2009), provided by (Tian et al., 2020) with 50 samples per class. This smaller subset enables us to conduct methods that have a higher computational cost for a fair comparison. To evaluate robustness, we conduct experiments on five distribution-shift datasets: ImageNetV2 (Shankar et al., 2020), ImageNet-R (Hendrycks et al., 2021a), ImageNet-Sketch (Wang et al., 2019), ObjectNet (Barbu et al., 2019), and ImageNet-A (Hendrycks et al., 2021b). For these datasets, we evaluate on subsets of 100 classes, except for ObjectNet, where we use the 113 classes overlapping with ImageNet. To accelerate evaluation, we use 10 samples per class. Top-1 accuracy and per-image computation cost (in GFLOPs) are reported as the performance and efficiency metrics, respectively.

**Baselines and Implementation Details.** In this work, we compare the proposed method against both discriminative and generative classifiers. We focus on models trained on ImageNet to conduct a fair comparison. For discriminative classifiers, we include ResNet-18, ResNet-34, ResNet-50, and ResNet-101 (He et al., 2016), as well as ViT-L/32, ViT-L/16, and ViT-B/16 (Dosovitskiy et al., 2020). For generative classifiers, we evaluate IBINN (Mackowiak et al., 2021), a normalizing flow-based generative classifier, and the diffusion classifier (Li et al., 2023). For IBINN, we use the model trained with $\beta = 1$. For diffusion-based generative classifiers, we examine two families: diffusion and rectified flow. In the diffusion case, we follow the setup of (Li et al., 2023) (DC) and adopt DiT-XL/2 (Peebles & Xie, 2023) at resolution 256 as the backbone. We report results under two settings: (i) ELBO estimation using 25 samples, and (ii) the two-stage approach proposed in the original paper, where 25 samples are used to select the top-5 candidates followed by 250 samples for refined prediction. For the rectified flow-based implementation, we report results using MeanFlow (Geng et al., 2025) (DC-MF) with a SiT/XL-2 backbone, which enables evaluating whether improved sampling efficiency translates into better classification performance. In this case, the noise prediction error in Eq. 3 is replaced with the velocity prediction error associated with the rectified flow formulation, and 25 samples are used for error estimation. For A-VARC$^+$, we use VAR-d16 at resolution 256 as the backbone and adopt a three-stage procedure. In the first stage, we use $\hat{p}_{\theta,6}(x|y)$ to narrow down the candidates to the top 10. In the second stage, we use $p_\theta(x|y)$ to select the top 3 candidates. In the final stage, we apply $\tilde{p}_{\theta,3}(x|y)$ with $\sigma = 0.1$ for the final prediction. Please refer to Sec. E in Appendix for more details.

**Quantitative Results.** Table 1 summarizes the comparison results. On ImageNet-100, the proposed A-VARC$^+$ improves both the accuracy and efficiency compared to the naive implementation VARC and achieves accuracy comparable to that of the 2-stage DiT based diffusion classifier, with less than a 1% drop, while requiring $89\times$ less computational cost. The efficiency gain primarily arises from the tractable likelihood and the candidate pruning strategy, whereas the enhanced accuracy can be attributed to likelihood smoothing and CCA finetuning. By contrast, IBINN attains the highest efficiency by modeling class-conditional likelihoods with a Gaussian Mixture Model, which enables fast classification via cluster distance computation but leads to substantially lower accuracy. It is worth noting that although rectified-flow models such as MeanFlow exhibit superior sampling efficiency for image generation compared to diffusion models, this advantage does not translate to improved performance in rectified flow-based diffusion classifiers. When evaluated with the same number of samples for error estimation, DC-MF performs significantly worse than the DiT-based counterpart. Our analysis suggests that MeanFlow suffers from higher prediction error compared to the DiT-based diffusion classifier. A possible explanation is that rectified-flow models are trained to approximate marginal velocity fields using supervision from conditional flows, as discussed in (Geng et al., 2025). This training mismatch may introduce additional noise into the error estimation, weakening the class-conditional signal and ultimately degrading classification performance.

In terms of robustness, consistent with the findings of (Li et al., 2023), generative classifiers exhibit improved robustness to adversarial shifts in ImageNet-A compared to ResNet-based models. However, for other distribution-shift datasets, the VAR classifier does not demonstrate any noticeable advantage. This suggests that the robustness property reported in diffusion-based methods (Jaini et al., 2023; Li et al., 2024) does not generalize to VAR. Interestingly, the DiT-based diffusion classifier significantly outperforms all discriminative models except ViT-L/32 on ImageNet-Sketch, highlighting its robustness to shifts from natural images to sketches. Neither IBINN nor A-VARC$^+$ exhibits this behavior, which implies that the observed robustness likely originates from the denoising training paradigm of diffusion models rather than from the generative objective itself. Finally, on ImageNet and its variants—including ImageNet-V2, ImageNet-R, and ObjectNet—discriminative models continue to achieve superior overall performance. This persistent gap highlights that gener-

ative classifiers remain an underexplored direction with considerable room for improvement. Nevertheless, given the rapid advances in generative modeling, we expect that generative classifiers will benefit from these developments and gradually narrow this gap in the near future.

Table 1: Comparison on ImageNet and across multiple distribution shifts.

| Method | ImageNet | | Distribution shifts | | | | |
|---|---|---|---|---|---|---|---|
| | Top-1 | GFLOPs | IN-V2 | IN-R | IN-Sketch | ObjectNet | IN-A |
| ResNet18 | 88.44 | 1.8 | 79.1 | 41.2 | 43.8 | 26.02 | 3.6 |
| ResNet34 | 89.96 | 3.7 | 81.3 | 40.9 | 46.1 | 30.62 | 5.1 |
| ResNet50 | 91.90 | 4.1 | 83.4 | 44.0 | 45.5 | 34.69 | 2.0 |
| ResNet101 | 92.14 | 7.8 | 84.8 | 44.6 | 50.3 | 36.99 | 6.8 |
| ViT-L/32 | 91.92 | 15.3 | 83.9 | 51.3 | 55.2 | 30.44 | 14.4 |
| ViT-L/16 | 93.22 | 59.7 | 86.0 | 49.4 | 49.7 | 33.98 | 18.0 |
| ViT-B/16 | 94.20 | 16.9 | 86.9 | 52.5 | 52.0 | 36.28 | 21.7 |
| IBINN | 51.12 | **9.2** | 40.9 | 13.2 | 14.6 | 3.98 | 3.2 |
| DC-MF$_{(25)}$ | 50.30 | 296861.3 | 44.0 | 4.5 | 3.9 | 10.27 | 4.8 |
| DC$_{(25)}$ | 86.32 | 286287.6 | 75.8 | 33.1 | 51.6 | 22.65 | 13.7 |
| DC$_{(25,250)}$ | **90.30** | 415056.0 | **80.6** | **38.3** | **53.7** | **29.38** | **16.2** |
| VARC | 83.30 | 14105.0 | 71.9 | 30.6 | 36.0 | 19.47 | 10.3 |
| A-VARC$^+$ | 89.32 | 4649.4 | 79.3 | 33.1 | 34.0 | 24.51 | 10.0 |

**Ablation Study.** Table 2 presents the ablation study of the two accuracy enhancement techniques adopted by A-VARC$^+$: likelihood smoothing and CCA finetuning. To focus on the analysis of the accuracy gain, the partial-scale candidate pruning technique is not applied in this experiment. Likelihood smoothing is applied only to the top-10 candidate classes, selected based on the class-conditional likelihood $p_\theta(x|y)$ from a standard forward pass. We use 10 samples for smoothing, as additional samples yield diminishing returns. The results show that likelihood smoothing consistently improves the performance of the baseline for all datasets, though at the cost of increased computation. In contrast, CCA finetuning enhances in-domain accuracy on ImageNet and closely related datasets such as ImageNetV2, ImageNet-R, and ObjectNet, but slightly reduces performance on ImageNet-A and ImageNet-Sketch. This suggests that CCA finetuning encourages the model to emphasize class-specific information, thereby improving discrimination within the training distribution but reducing generalization to larger distribution shifts.

Table 2: Ablation study on likelihood smoothing and CCA finetuning.

| Smooth ($S$=10) | CCA | ImageNet | | Distribution shifts | | | | |
|---|---|---|---|---|---|---|---|---|
| | | Top-1 | GFLOPs | IN-V2 | IN-R | IN-Sketch | ObjectNet | IN-A |
| | | 83.30 | 14105.0 | 71.9 | 30.6 | 36.0 | 19.47 | 10.3 |
| ✓ | | 88.26 | 28210.0 | 77.1 | 33.6 | **40.4** | 24.78 | **11.0** |
| | ✓ | 88.68 | 14105.0 | 80.3 | **34.5** | 34.8 | 25.75 | 9.9 |
| ✓ | ✓ | **89.72** | 28210.0 | **81.2** | 33.9 | 36.0 | **26.73** | 10.9 |

# 6 INTRIGUING PROPERTIES

In this section, we discuss the intriguing properties of the VAR-based classifier that distinguish it from conventional discriminative classifiers.

## 6.1 VISUAL EXPLAINABILITY

The tractable likelihood of the VAR model inherently provides visual explainability. The concept of pointwise mutual information (PMI), defined as $\log \frac{p(x|y)}{p(x)}$, has been widely used in NLP tasks (Church & Hanks, 1990; Levy & Goldberg, 2014) to measure word associations. Here, we show that this concept can be naturally extended by the VAR-based classifier to provide visual explanations.

The goal of visual explanation is to capture fine-grained associations between local image regions and a target label $y$, thereby clarifying why the model makes a particular decision. Since autoregressive models compute token-wise likelihoods, we can extend pointwise mutual information to token-wise mutual information (TMI), which measures the association between a token and a label as follows:

$$\log \frac{p_\theta(r_k^{(i,j)} \mid r_1, r_2, \cdots, r_{k-1}, y)}{p_\theta(r_k^{(i,j)} \mid r_1, r_2, \cdots, r_{k-1})} \tag{12}$$

where $r_k^{(i,j)}$ denotes the $(i,j)$-th token of the $k$-th scale token map. This ratio can be obtained for each token with only two forward passes. Moreover, this concept can be extended to contrastive explanations, which highlight why a prediction is made in favor of one class over another. Fig. 3 illustrates that token-wise mutual information effectively identifies regions strongly associated with the label "little blue heron", thereby revealing the basis of the model's prediction. It also provides contrastive evidence by explaining why the image is classified as a "little blue heron" rather than a "goose". This offers direct and interpretable insight into the decision-making process of the VAR classifier. Please refer to Fig. 7 and Fig. 8 in the Appendix for more visualization results.

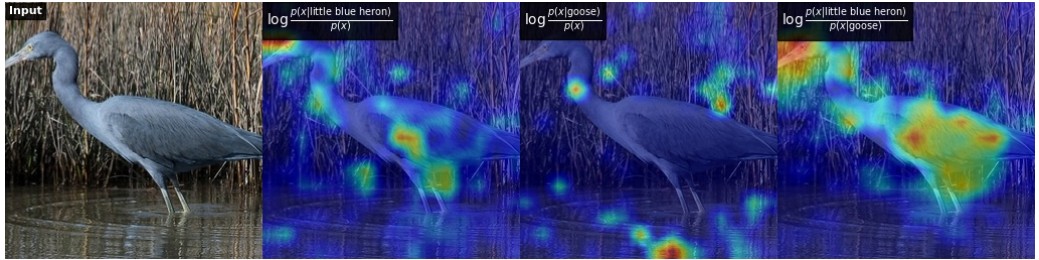

Figure 3: Visual explanation of A-VARC$^+$ using TMI. From left to right: the input image, TMI conditioned on the true label, TMI conditioned on the highest-ranked incorrect label, and the contrastive explanation between them.

The token-wise mutual information can be viewed as an attribution method that produces an attribution score for each token. To evaluate attribution quality, we adopt the insertion and deletion metrics introduced in (Petsiuk et al., 2018), which are widely used in the explainability literature. Tokens are first sorted according to their attribution scores; then, they are gradually inserted or removed to measure the change in the predicted probability of the ground-truth class. The area under the curve (AUC) is used for evaluation. Intuitively, for the insertion metric, a higher AUC is preferred, as it indicates that tokens with high attribution scores meaningfully support the ground-truth class. For the deletion metric, a lower AUC is preferred, as removing highly attributed tokens should decrease the ground-truth probability. We additionally report LIME (Ribeiro et al., 2016) and SHAP (Lundberg & Lee, 2017) as baselines. Please refer to Sec. C in Appendix for more details.

Table 3 presents the quantitative analysis of the visual explanation methods. For A-VARC, LIME demonstrates the strongest overall performance, while TMI performs comparably to SHAP on the insertion metric and achieves the second-best score on the deletion metric. For A-VARC$^+$, TMI outperforms all other methods on both insertion and deletion metrics. These results are consistent with the qualitative observations in Fig. 6, where TMI becomes more focused on class-relevant regions after finetuning, leading to improved explainability.

Table 3: Quantitative analysis of visual explanation methods. The reported metrics correspond to the average area-under-the-curve (AUC) averaged across the ImageNet-100 dataset.

| | Metric | LIME | SHAP | TMI (Ours) |
|---|---|---|---|---|
| A-VARC | Insertion ($\uparrow$) | **0.979** | 0.853 | 0.845 |
| | Deletion ($\downarrow$) | **0.192** | 0.432 | 0.346 |
| A-VARC$^+$ | Insertion ($\uparrow$) | 0.939 | 0.902 | **0.944** |
| | Deletion ($\downarrow$) | 0.614 | 0.746 | **0.605** |

## 6.2 CLASS-INCREMENTAL LEARNING

Unlike discriminative classifiers, which rely on a unified softmax layer across all classes to obtain logits for classification, generative classifiers base their predictions solely on class-conditional likelihoods. Since these likelihoods can be learned independently for each class, generative classifiers naturally adapt to class-incremental learning without suffering from catastrophic forgetting, as noted in (Van De Ven et al., 2021). This provides a distinct advantage over discriminative models, which are vulnerable to catastrophic forgetting and typically require storing a portion of past data as "rehearsal" to preserve performance.

To investigate whether recent generative classifiers exhibit similar behavior, we conduct a proof-of-concept class-incremental learning experiment on the first 10 classes of ImageNet. The classes are partitioned into two tasks, each containing 5 classes. Discriminative classifiers are trained sequentially, first on Task 1 and then on Task 2. For generative classifiers, two separate models are independently trained on each task, following the setup in (Van De Ven et al., 2021). We evaluate both diffusion-based and VAR-based generative classifiers. Specifically, we use ResNet-50 as the discriminative baseline, DiT-S/2 for the diffusion classifier, and VAR-d8 for the VAR classifier. All models are trained from scratch for 1,000 epochs, except DiT-S/2, which is trained for 2,000 epochs. Afterward, the VAR model is further finetuned with CCA for an additional 10 epochs.

As shown in Table 4, without rehearsal data, the discriminative model suffers severe catastrophic forgetting. Although methods such as CWR (Lomonaco & Maltoni, 2017) can mitigate this issue, their reliance on fixed feature extractors limits performance on new tasks. In contrast, the generative classifiers, trained to model class-conditional likelihoods independently, adapts naturally to new tasks and achieves promising performance without requiring additional data or complex techniques. This provides a promising solution for creating a unified classifier by simply merging classifiers trained on different datasets, making it capable of recognizing an expanded set of classes without retraining. Compared to the VAE used in previous work Van De Ven et al. (2021), the VAR model, along with the techniques of A-VARC$^+$, provides a powerful alternative for future research in class-incremental learning.

Table 4: Class-incremental learning experiment on the first 10 classes of ImageNet.

| | None | | | CWR | | | DC | | | A-VARC$^+$ | |
| Task1 | Task2 | Avg | Task1 | Task2 | Avg | Task1 | Task2 | Avg | Task1 | Task2 | Avg |
|---|---|---|---|---|---|---|---|---|---|---|---|
| 0.0 | 82.4 | 41.2 | 83.2 | 61.6 | 72.4 | 78.4 | 73.6 | 76.0 | 72.4 | 82.4 | **77.4** |

## 7 CONCLUSION

In this work, we investigate VAR-based generative classifiers and propose A-VARC$^+$, which further improves both accuracy and efficiency, achieving performance comparable to DiT-based diffusion classifiers while requiring substantially less computational cost. Although our analysis indicates that VAR-based classifiers do not inherit certain properties exhibited by diffusion-based models, such as robustness to distribution shift, we uncover other notable characteristics, including visual explainability and natural adaptability to class-incremental learning. These findings offer a deeper understanding of the strengths and limitations of generative classifiers and point toward promising directions for future research.

## REPRODUCIBILITY STATEMENT

To ensure reproducibility, we release our implementation at `https://github.com/Yi-Chung-Chen/A-VARC`, along with detailed instructions in Sec. F of the Appendix for obtaining the evaluation subsets used in our experiments. We also provide the pseudo-code of A-VARC in Algorithm 1. For discriminative baselines, pretrained models are publicly available in

`torchvision`[1]. The code and pretrained weights for IBINN[2], DC[3], VAR[4], and CCA[5] are also accessible via their respective official repositories.

ACKNOWLEDGEMENTS

This work is supported in part by the US National Science Foundation under grants NSF IIS-2141037 and IIS-2212097, the US Army Research Laboratory under award W911NF-2020-221, and the Office of Naval Research under contract N00014-23-C-1016. Any opinions, findings, and conclusions or recommendations expressed in this material are those of the author(s) and do not necessarily reflect the views of the sponsors.

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

# A USE OF LARGE LANGUAGE MODELS

In this work, we made limited use of large language models (LLMs) to assist with writing and reference search. Specifically, LLMs were used to polish the text for clarity and readability, and to conduct preliminary surveys for identifying relevant references. All generated content was carefully reviewed and edited by the authors to ensure that it faithfully reflects the intended meaning. Likewise, all references retrieved through LLM-assisted searches were manually verified for accuracy and alignment with the described content.

# B ADDITIONAL ABLATION STUDY

## B.1 LIKELIHOOD SMOOTHING WITH VARYING NUMBERS OF SAMPLES

Following the setting in Table 2, we provide an additional ablation study using different values of $S$ for likelihood smoothing in Table 5. For A-VARC, the accuracy increases as $S$ grows and saturates at $S = 16$, yielding an overall improvement of 5.12%. For A-VARC$^+$, saturation occurs earlier at $S = 8$ with a smaller improvement of 1.06%, indicating that the gain from smoothing is reduced when combined with CCA. In both cases, the results consistently show that applying likelihood smoothing improves classification accuracy.

Table 5: Ablation study of likelihood smoothing with varying numbers of samples.

|  | $S$=1 | $S$=2 | $S$=4 | $S$=8 | $S$=10 | $S$=16 | $S$=32 |
|---|---|---|---|---|---|---|---|
| A-VARC | 83.30 | 85.30 | 87.18 | 88.18 | 88.26 | 88.42 | 88.42 |
| A-VARC$^+$ | 88.68 | 89.42 | 89.30 | 89.74 | 89.72 | 89.54 | 89.72 |

## B.2 VARIANCE FOR LIKELIHOOD SMOOTHING

Likelihood smoothing averages the likelihoods of neighboring samples in the latent space to promote local smoothness. The neighborhood size is controlled by the variance parameter $\sigma$. Table 6 reports an ablation study of A-VARC over different $\sigma$ values. A small $\sigma$ limits smoothing to a narrow neighborhood, while a large $\sigma$ may undesirably average over semantically dissimilar samples. Empirically, we find that $\sigma = 0.1$ provides the most favorable results.

Table 6: Effect of varying the likelihood smoothing variance $\sigma$.

| $\sigma$ | 0.01 | 0.05 | 0.1 | 0.5 | 1.0 |
|---|---|---|---|---|---|
| Acc | 85.50 | 87.72 | 88.26 | 84.38 | 39.54 |

## B.3 PARTIAL-SCALE CANDIDATE PRUNING WITH LIKELIHOOD SMOOTHING

We explored using more samples during the candidate pruning stage by combining likelihood smoothing with partial-scale pruning. The results are provided in Fig. 4. It shows that although using more samples can slightly improve the top-10 accuracy, it substantially increases computational cost. Therefore, we recommend using $S = 1$ at this stage to achieve better efficiency.

# C VISUAL EXPLAINABILITY

## C.1 IMPLEMENTATION DETAILS OF QUANTITATIVE ANALYSIS

To compute attribution scores, LIME (Ribeiro et al., 2016) and SHAP (Lundberg & Lee, 2017) perturb the features of interest and measure the corresponding variation in the target function. In our setting, we apply these explanation methods to estimate an attribution score for each token, using the predicted probability of the ground-truth class as the target function. For each perturbation, a binary mask vector $M = \{m_k^{(i,j)}\}$ is used to select a subset of tokens. The predicted probability of

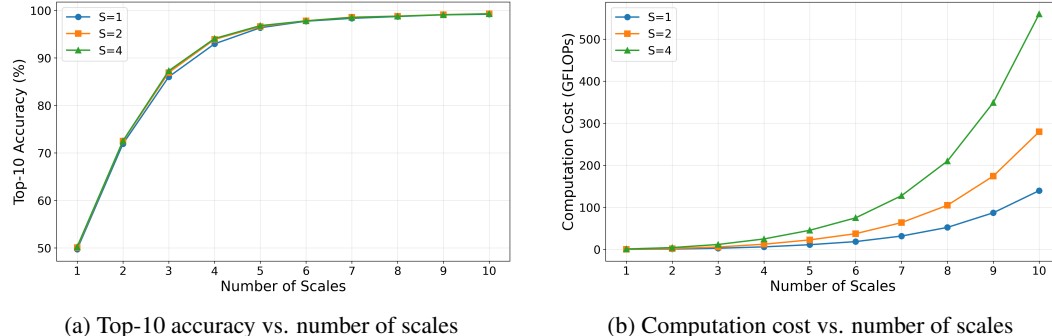

(a) Top-10 accuracy vs. number of scales

(b) Computation cost vs. number of scales

Figure 4: Comparison of top-10 accuracy and computational cost across different numbers of scales for varying values of $S$.

the ground-truth class, $p_\theta(y_{gt} \mid x, M)$, is computed as follows:

$$l_\theta(x \mid y, M) = \sum_{(i,j,k)} m_k^{(i,j)} \log p_\theta(r_k^{(i,j)} \mid r_1, r_2, \cdots, r_{k-1}, y) \tag{13}$$

$$p_\theta(y_{gt} \mid x, M) = \frac{\exp(l_\theta(x \mid y_{gt}, M))}{\sum_i \exp(l_\theta(x \mid y_i, M))} \tag{14}$$

where $m_k^{(i,j)} \in \{0, 1\}$ is a binary indicator determining token inclusion, and $l_\theta(x \mid y, M)$ denotes the mask-aware log-likelihood of $x$. LIME then fits a linear model on the perturbation outputs to obtain attribution scores, whereas SHAP uses Kernel SHAP to approximate Shapley values. For both methods, we use 5,000 perturbations during evaluation. The resulting attribution scores are then evaluated using the insertion and deletion metrics. An example AUC curve for these metrics is shown in Fig. 5.

Note that we precompute the token-wise log-likelihoods and apply the masking strategy in Eq. 13 to approximate the effect of token insertion or deletion. This allows efficient computation of token attributions while avoiding potential out-of-distribution artifacts that may arise from directly altering the sequence.

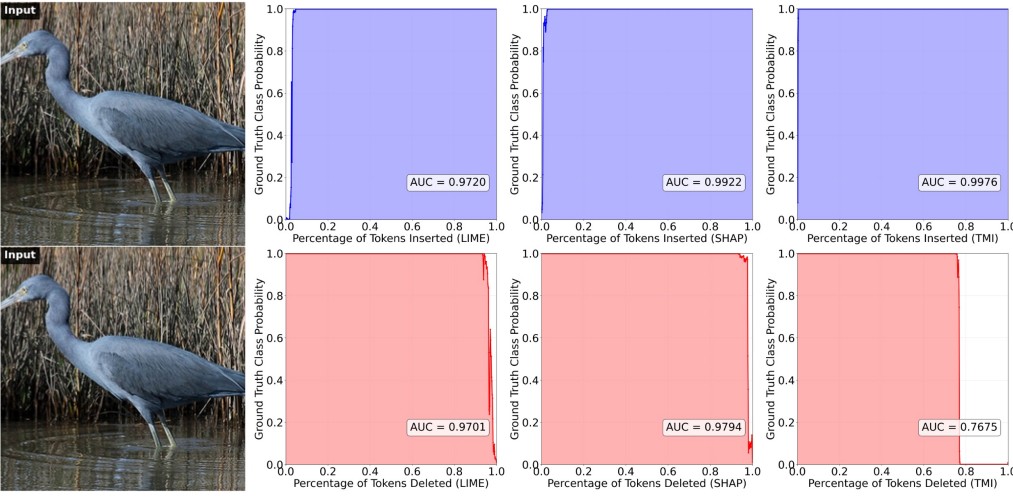

Figure 5: Insertion/deletion analysis of explanation methods. The first row shows insertion results, and the second row shows deletion results. Methods from left to right are LIME, SHAP, and TMI.

## C.2 VISUALIZATION RESULTS

In this section, we provide additional visual explanations of the VAR-based classifier. Fig. 7 and Fig. 8 illustrate that the classification results are indeed driven by regions corresponding to the foreground object. For example, in the second row of Fig. 7, the example is correctly classified as a green mamba, not because of the background green leaf, but because the model focuses on the snake's head.

We also analyze failure cases in Fig. 9. The most common type of error occurs when the model fails to distinguish between visually similar classes, such as hen vs. cock or chihuahua vs. toy terrier. Even when the model correctly identifies important regions for each class, it may still produce an incorrect final prediction. Another common error arises in scenarios where multiple candidate objects appear in the same image. For instance, the ground-truth label of the third-row example is 'modem', but the presence of a laptop in the same image misleads the classifier into predicting 'laptop'. A similar issue is observed in the last example, where the co-occurrence of a tabby cat and a bassinet results in an incorrect prediction.

These examples demonstrate that visual explanations can provide valuable insights into the decision-making process of the VAR-based classifier, enabling developers to better understand model behavior and make informed adjustments during development.

## D TRADE-OFF BETWEEN GENERATIVE AND DISCRIMINATIVE PERFORMANCE.

The trade-off between generative and discriminative performance in conditional generative classifiers has been discussed in prior work (Fetaya et al., 2020). While this phenomenon is less pronounced in the VAR-d16 model, it becomes increasingly evident as model size grows. Table 7 reports the performance of VAR classifiers with different model sizes, including accuracy on ImageNet-100 and the FID reported by Chen et al. (2024b) (without likelihood smoothing or candidate pruning). As model size increases, generative performance improves, as reflected by lower FID, but classification accuracy on ImageNet-100 drops substantially.

This degradation can be attributed to the dilution of class-conditional information by structural information. Specifically, the class-conditional likelihood of each token depends on both class and structural information. Larger models, with stronger generative capacity, are able to more accurately infer tokens at subsequent scales even when conditioned on an incorrect class label. This is evidenced by the simultaneous increase in both $p(x|y)$ and $p(x|y^{neg})$ as model size grows. As a result, the contribution of class information to likelihood estimation diminishes, leading to weaker discriminative ability. As illustrated in Fig. 10, larger models increasingly fail to distinguish between visually similar classes, such as cock vs. hen or great white shark vs. tiger shark.

To address this issue, one promising direction is to improve the training objective so that it more effectively preserves class information. While recent advances such as CCA represent a meaningful step in this direction, our results indicate that CCA alone is insufficient to fully resolve the problem. Another complementary direction is to disentangle class information from structural information, thereby preventing the dilution effect associated with increased likelihood. We leave the exploration of these directions to future work.

Table 7: Trade-off between generative and discriminative performance of the VAR classifier across different model sizes.

|  | d16 | | d20 | | d24 | | d30 | |
|  | VAR | CCA | VAR | CCA | VAR | CCA | VAR | CCA |
|---|---|---|---|---|---|---|---|---|
| Accuracy | 83.30 | 88.68 | 80.80 | 88.90 | 75.68 | 84.92 | 64.96 | 71.64 |
| FID | 12.00 | 4.03 | 8.48 | 3.02 | 6.20 | 2.63 | 5.26 | 2.54 |

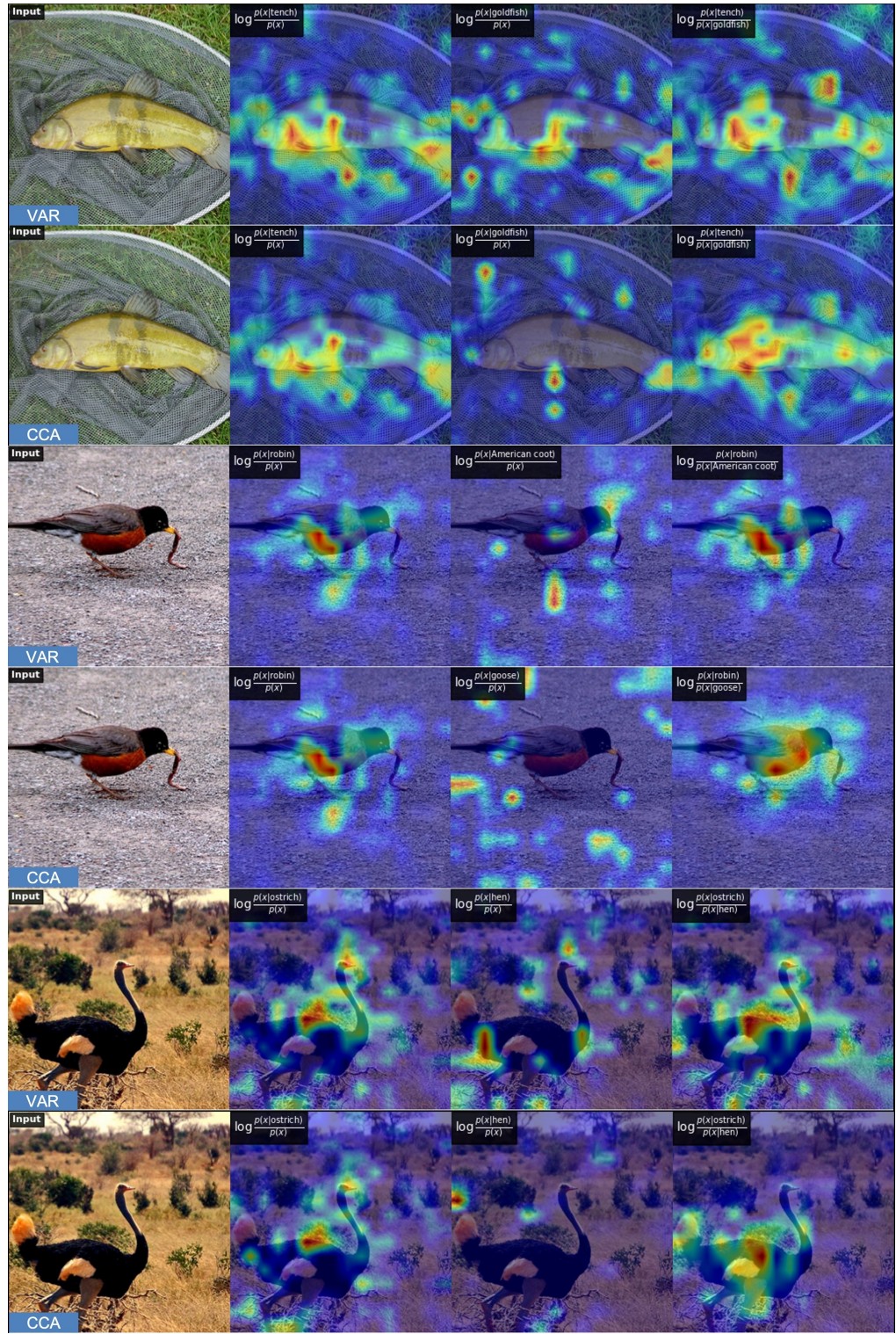

Figure 6: Impact of CCA finetuning. From left to right: the input image, TMI conditioned on the true label, TMI conditioned on the highest-ranked incorrect label, and the contrastive explanation between them.

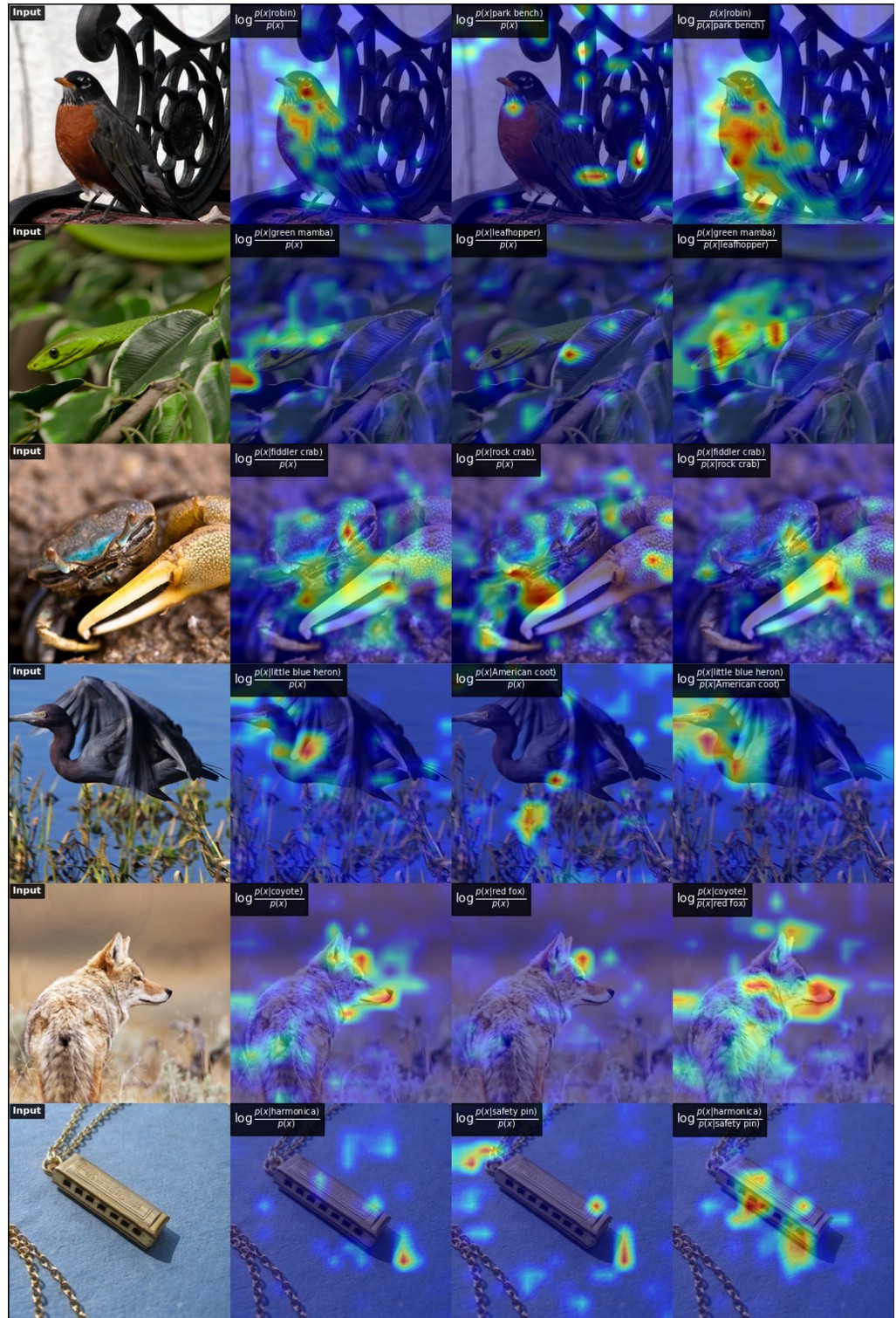

Figure 7: Visual explanation of A-VARC$^+$ using TMI. From left to right: the input image, TMI conditioned on the true label, TMI conditioned on the highest-ranked incorrect label, and the contrastive explanation between them.

# E    PSUEDO CODE

Algorithm 1 outlines the classification procedure of the proposed Adaptive VAR Classifier (A-VARC). The likelihood estimation strategy is composed of three forms of likelihood estimation:

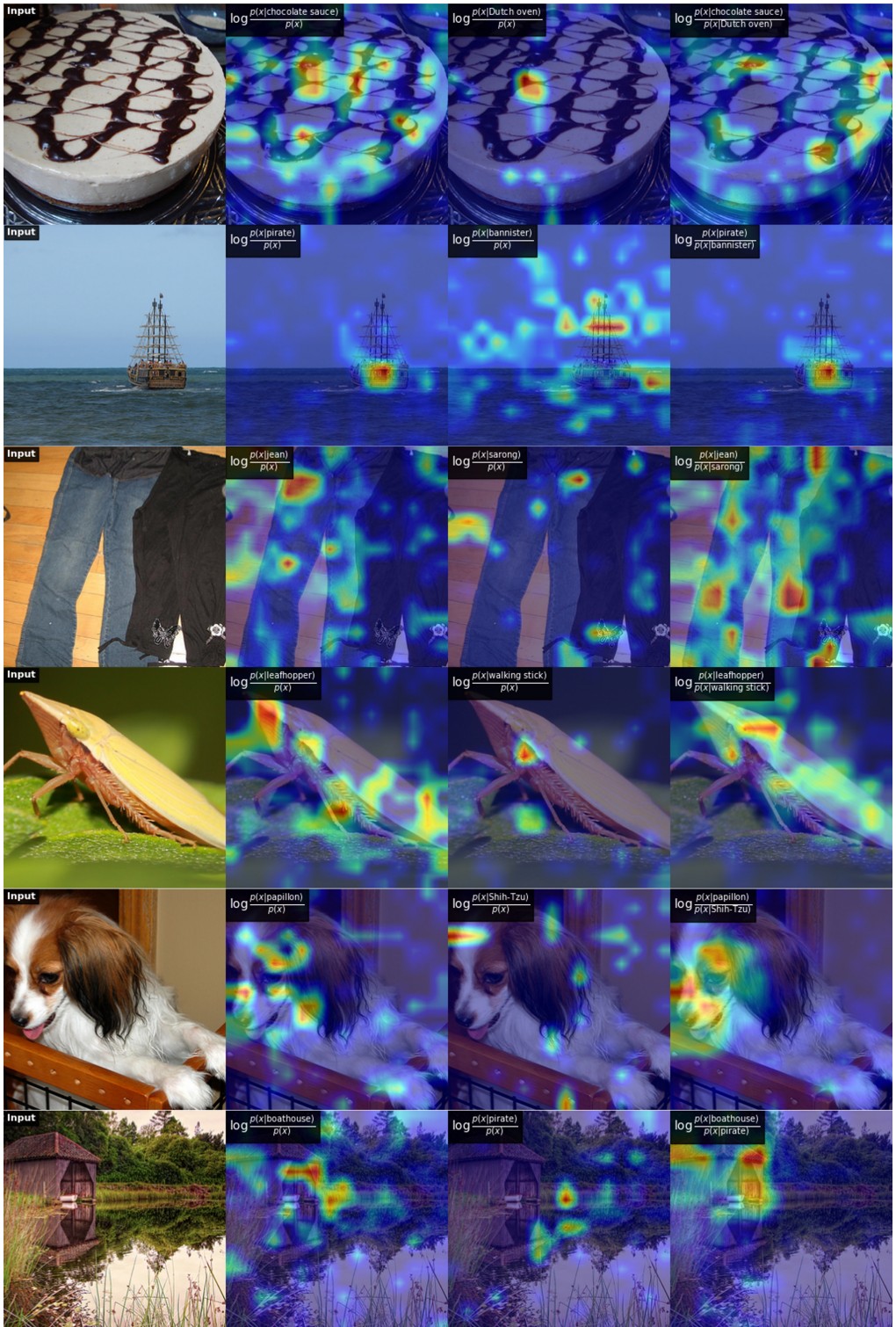

Figure 8: Visual explanation of A-VARC$^+$ using TMI. From left to right: the input image, TMI conditioned on the true label, TMI conditioned on the highest-ranked incorrect label, and the contrastive explanation between them.

$p_\theta(x|y)$ (Eq. 7), $\tilde{p}_{\theta,S}(x|y)$ (Eq. 9), and $\hat{p}_{\theta,K'}(x|y)$ (Eq. 10). This flexible design enables a wide

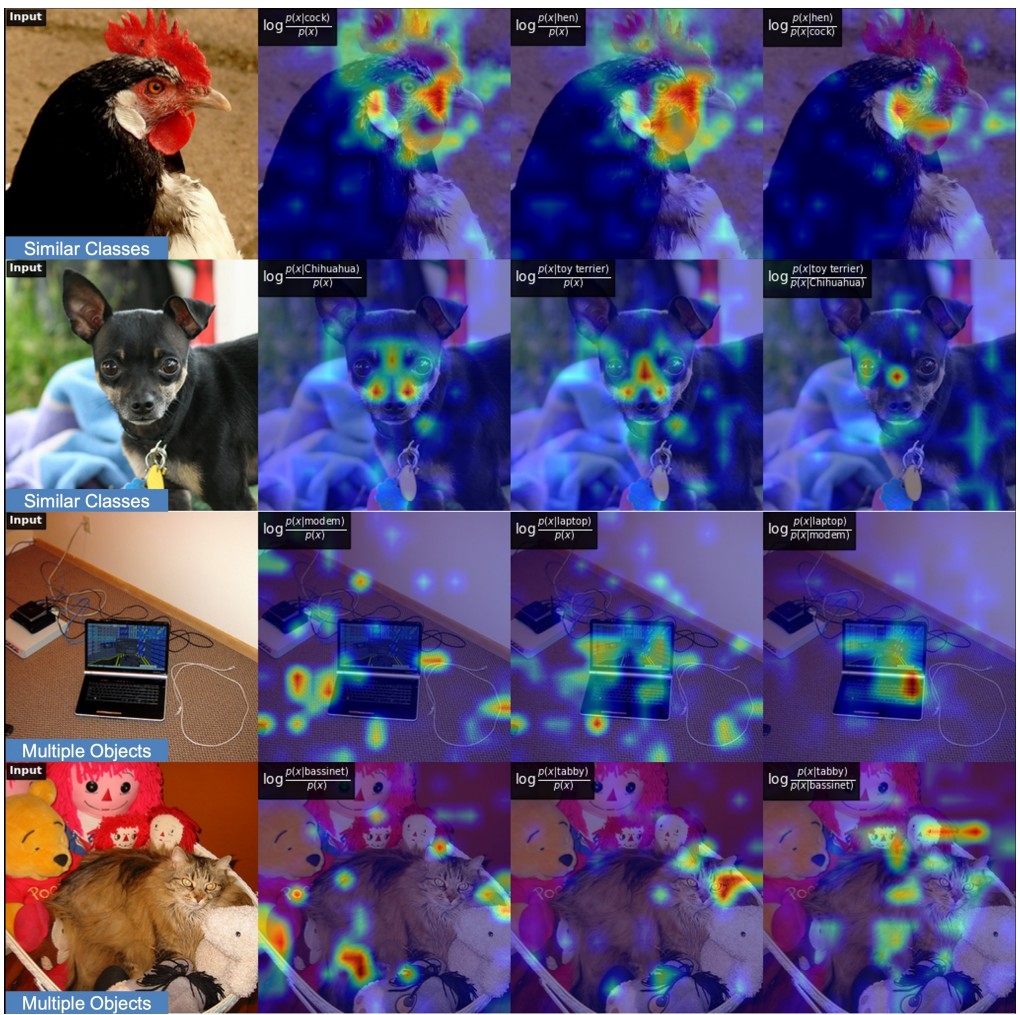

Figure 9: Visual explanation for failure cases. From left to right: the input image, TMI conditioned on the true label, TMI conditioned on the highest-ranked incorrect label, and the contrastive explanation between them.

range of combinations, allowing users to balance accuracy and efficiency according to their requirements. For the experiments reported in Table1, we adopt a three-stage configuration with $N_{stage} = 3$, $\mathcal{K} = (10, 3, 1)$ and $\mathcal{M} = (\hat{p}_{\theta,6}(x|y), p_\theta(x|y), \tilde{p}_{\theta,3}(x|y))$. Note that, while the algorithm is presented using likelihoods for clarity, in practice, we compute log-likelihoods to ensure numerical stability.

## F    IMPLEMENTATION DETAILS

To enhance reproducibility, we provide details of the subsets used in Table 1. For ImageNet, ImageNet-V2, and ImageNet-Sketch, we adopt the same set of classes provided by Tian et al. (2020), as listed in Table 8. Since ImageNet-A and ImageNet-R do not include all classes from ImageNet-100, we select the overlapping classes and list them in Table 9 and Table 10, respectively. For ObjectNet, we use all overlapping classes reported in Barbu et al. (2019), with implementation support from the diffusion classifier's (Li et al., 2023) repository[6].

To reduce evaluation cost on distribution-shift datasets, we further subsample 10 samples per class. For most datasets, we sort file names alphabetically and select the first 10 samples per class. For

---

[6]https://github.com/diffusion-classifier/diffusion-classifier

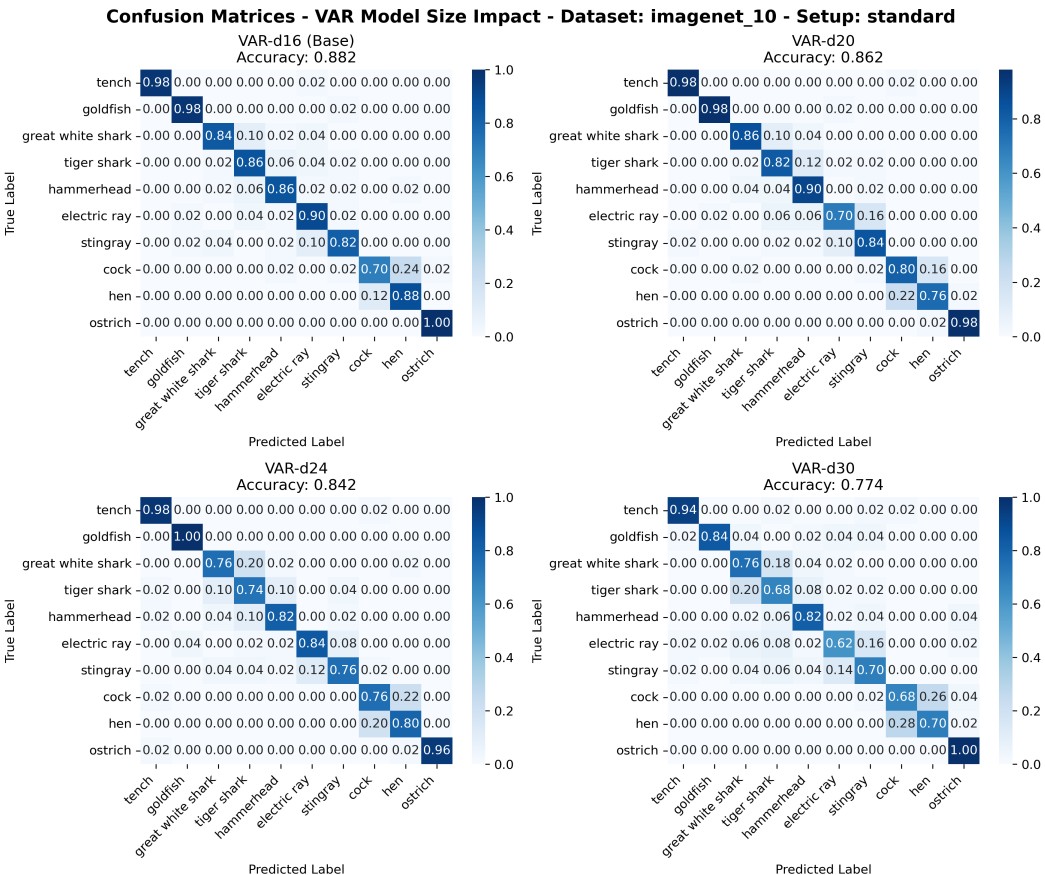

Figure 10: Confusion matrices of the VAR classifier evaluated on the first 10 classes of ImageNet.

---

**Algorithm 1** Adaptive VAR Classifier (A-VARC)

---

**Require:** Test image $x$; Initial set of candidate labels $\mathcal{Y} = \{y_i\}_{i=1}^n$; Number of stages $N_{\text{stages}}$; Sequence of candidate counts $\mathcal{K} = (k_1, \ldots, k_{N_{\text{stages}}})$; Likelihood estimation strategy $\mathcal{M} = (m_1, \ldots, m_{N_{\text{stages}}})$.

1: Initialize candidate set $C_0 \leftarrow \mathcal{Y}$
2: Initialize a map $L$ to store likelihood scores
3: **for** stage $i = 1, \ldots, N_{\text{stages}}$ **do**
4:     **for** each candidate $y_j \in C_{i-1}$ **do**
5:         $L(y_j) \leftarrow \text{ComputeLikelihood}(\mathbf{x}, y_j, \text{method} = m_i)$     ▷ e.g., computing $\hat{p}_{\theta, K'}(x \mid y_j)$
6:     **end for**
7:     Let $C'_{i-1}$ be the set $C_{i-1}$ sorted by descending scores $L(\cdot)$
8:     $C_i \leftarrow$ the first $k_i$ elements of $C'_{i-1}$     ▷ Prune candidates with the lowest scores
9: **end for**
10: **return** $\arg\max_{y \in C_{N_{\text{stages}}}} L(y)$

---

ImageNet-R, which includes multiple styles (e.g., art, cartoon, deviantart), we first sort styles alphabetically and then apply a round-robin ordering across styles (e.g., art_0.jpg, cartoon_0.jpg, deviantart_0.jpg, ...), ensuring that as many styles as possible are represented.

Table 8: List of ImageNet 100 classes used in our experiments, identified by their WordNet IDs (n-numbers).

| List of ImageNet 100 classes | | | | | | |
|---|---|---|---|---|---|---|
| n02869837 | n01749939 | n02488291 | n02107142 | n13037406 | n02091831 | n04517823 |
| n04589890 | n03062245 | n01773797 | n01735189 | n07831146 | n07753275 | n03085013 |
| n04485082 | n02105505 | n01983481 | n02788148 | n03530642 | n04435653 | n02086910 |
| n02859443 | n13040303 | n03594734 | n02085620 | n02099849 | n01558993 | n04493381 |
| n02109047 | n04111531 | n02877765 | n04429376 | n02009229 | n01978455 | n02106550 |
| n01820546 | n01692333 | n07714571 | n02974003 | n02114855 | n03785016 | n03764736 |
| n03775546 | n02087046 | n07836838 | n04099969 | n04592741 | n03891251 | n02701002 |
| n03379051 | n02259212 | n07715103 | n03947888 | n04026417 | n02326432 | n03637318 |
| n01980166 | n02113799 | n02086240 | n03903868 | n02483362 | n04127249 | n02089973 |
| n03017168 | n02093428 | n02804414 | n02396427 | n04418357 | n02172182 | n01729322 |
| n02113978 | n03787032 | n02089867 | n02119022 | n03777754 | n04238763 | n02231487 |
| n03032252 | n02138441 | n02104029 | n03837869 | n03494278 | n04136333 | n03794056 |
| n03492542 | n02018207 | n04067472 | n03930630 | n03584829 | n02123045 | n04229816 |
| n02100583 | n03642806 | n04336792 | n03259280 | n02116738 | n02108089 | n03424325 |
| n01855672 | n02090622 | | | | | |

Table 9: List of ImageNet-A 100 classes used in our experiments, identified by their WordNet IDs (n-numbers).

| List of ImageNet-A 100 classes | | | | | | |
|---|---|---|---|---|---|---|
| n01531178 | n01580077 | n01616318 | n01631663 | n01641577 | n01669191 | n01677366 |
| n01687978 | n01694178 | n01774750 | n01820546 | n01833805 | n01843383 | n01847000 |
| n01855672 | n01910747 | n01924916 | n01944390 | n01986214 | n02051845 | n02077923 |
| n02099601 | n02106662 | n02110958 | n02119022 | n02133161 | n02137549 | n02165456 |
| n02174001 | n02190166 | n02206856 | n02219486 | n02236044 | n02259212 | n02268443 |
| n02279972 | n02280649 | n02325366 | n02445715 | n02454379 | n02504458 | n02655020 |
| n02730930 | n02782093 | n02802426 | n02814860 | n02879718 | n02883205 | n02895154 |
| n02906734 | n02948072 | n02951358 | n02999410 | n03014705 | n03026506 | n03223299 |
| n03250847 | n03255030 | n03355925 | n03444034 | n03452741 | n03483316 | n03590841 |
| n03594945 | n03617480 | n03666591 | n03720891 | n03721384 | n03788195 | n03888257 |
| n04033901 | n04099969 | n04118538 | n04133789 | n04146614 | n04147183 | n04179913 |
| n04252077 | n04252225 | n04317175 | n04366367 | n04376876 | n04399382 | n04442312 |
| n04456115 | n04507155 | n04509417 | n04591713 | n07583066 | n07697313 | n07697537 |
| n07714990 | n07718472 | n07734744 | n07768694 | n07831146 | n09229709 | n11879895 |
| n12144580 | n12267677 | | | | | |

Table 10: List of ImageNet-R 100 classes used in our experiments, identified by their WordNet IDs (n-numbers).

| List of ImageNet-R 100 classes | | | | | | |
|---|---|---|---|---|---|---|
| n01484850 | n01514859 | n01531178 | n01534433 | n01614925 | n01616318 | n01632777 |
| n01774750 | n01820546 | n01833805 | n01843383 | n01847000 | n01855672 | n01860187 |
| n01882714 | n01944390 | n01983481 | n02007558 | n02056570 | n02066245 | n02086240 |
| n02088094 | n02088238 | n02096585 | n02097298 | n02098286 | n02102318 | n02106166 |
| n02106550 | n02106662 | n02108089 | n02108915 | n02110341 | n02113624 | n02113799 |
| n02117135 | n02119022 | n02128757 | n02129165 | n02130308 | n02190166 | n02206856 |
| n02236044 | n02268443 | n02279972 | n02317335 | n02325366 | n02346627 | n02356798 |
| n02363005 | n02364673 | n02395406 | n02398521 | n02410509 | n02423022 | n02486410 |
| n02510455 | n02749479 | n02793495 | n02797295 | n02808440 | n02814860 | n02883205 |
| n02939185 | n02950826 | n02966193 | n02980441 | n03124170 | n03372029 | n03424325 |
| n03452741 | n03481172 | n03495258 | n03630383 | n03676483 | n03710193 | n03773504 |
| n03775071 | n03930630 | n04118538 | n04254680 | n04266014 | n04310018 | n04347754 |
| n04389033 | n04522168 | n04536866 | n07693725 | n07697313 | n07697537 | n07714571 |
| n07714990 | n07720875 | n07745940 | n07749582 | n07753275 | n07753592 | n09835506 |
| n10565667 | n12267677 | | | | | |

