# OpenReview forum: "Your VAR Model is Secretly an Efficient and Explainable Generative Classifier"
_ICLR.cc/2026/Conference — ICLR 2026 Poster_

### Official Review · Reviewer_1Nsc · 2025-10-25

**Soundness:** 3
**Presentation:** 3
**Contribution:** 2
**Rating:** 4
**Confidence:** 4

**Summary:**

This paper leverages Vector Auto Regressive (VAR) models, a class of generative models to the classification task by applying Bayes rule.

**Strengths:**

Strengths of var:

- Faster (81x) and better quality compared to diffusion models  [Tian paper ]
- AR models have tractable likelihood

This paper, puts these strengths of the VAR models to use in the task of explainability and classification.

**Weaknesses:**

### There are multiple claims in the paper that are not validated.

**Superior Tradeoff / Pareto Dominance**
Let $a, b \in \mathbb{R}^2$ with objective values
$f(a) = (f_1(a), f_2(a))$ and $f(b) = (f_1(b), f_2(b))$.
We say that $a$ has a superior tradeoff to $b$
(or $a$ Pareto-dominates $b$), denoted $a \prec b$, if
$f_1(a) \le f_1(b), \quad f_2(a) \le f_2(b), \quad \text{and} \quad \exists j \in \{1,2\}: f_j(a) < f_j(b).$

However on the accuracy front VAR does not dominate. so at best you can say tradeoff and lies on Pareto front but cannot say Superior Tradeoff.

**Ablation on other candidate pruning**

Candidate pruning is a popular technique in diffusion classifier. There are multiple ways to prune, and more the pruning Figure 4.[https://arxiv.org/pdf/2303.16203] and more samples in the pruning process better the performance. You will get the same Figure 2 if you do uniform scale, the performance improves with number of selected scales. Can you provide experiments on various pruning techniques to validate the partial-scale hypothesis discussed in Section 4.2

**Ablation on Smoothness**

I agree with the authors that the adversarial noise, causes token drift. Is the model evaluated on Adversarial noise? No, So this section does not contribute to any new information. Do Authors perform ablation on S= [0, ….. N]? No. So How can authors conclude adding smoothness lead to improved performance?

**Unfair Comparison**

1. Class incremental learning can be seen in other generative models as well

    Class incremental learning is not just related VAR, but also observed in other types of generative models.  Consider the control net architecture in diffusion models, where you can add additional classes without even adding other model complexity. Can authors put to comparison how VAR model fares on class incremental learning in comparison to ControlNet?

2. DiT vs VAR
    1. Over the years there are multiple improvement to Diffusion classifier to address inference speed
        1. Rectified flow, few steps to clean image ( Instead of sampling 25 noise scales, you can get away with 5 noise scales. )
        2. Parallelisation, speed of  Diffusion classifier can be significantly improved by Parallelisation.  Pruning to 25 noise scales to an image and for an 100 class classification, you can pass everything in a 25 * 100 times batch and obtain the results quickly. ( time will be much faster ).

    I agree that even with improvements it is very difficult to beat VAR with respect to inference time, it would also be helpful to mention flops, should be a better metric.


**Strong Claims**

I am not sure what does “novel” generative classifier mean? All the works, prior to this method, such as diffusion classifier does not call them as a novel classifier. It is applying Bayes’ rule to generative models to perform classification task.

I don’t agree generative classifiers, particularly diffusion classifiers are not popular they are very popular on zero shot image classification tasks. In addition, diffusion models achieve better compositional generalisation [ https://arxiv.org/abs/2501.05707, https://arxiv.org/pdf/2503.04687]. However, using as a pure classifier, low adoption of generative classifiers can be attributed to the linear scalability with respect to number of classes, and not accurately modelling of $P(x|y)$

**Questions:**

Please refer to weakness section.

---

> ### Author Response · Authors · 2025-11-21
>
> We thank the reviewer for the insightful review and constructive comments. We provide our responses as follows.
>
> > Superior tradeoff
>
> Thank you for pointing out this point. We have updated the statement accordingly.
>
> > Ablation on other candidate pruning
>
> The candidate pruning strategy adopted by [f] relies on 25 samples to approximate the ELBO during the first-stage pruning, which requires 25 full forward passes. As shown in their Figure 2, the reconstruction error conditioned on the correct label, although smaller on average, is not consistently lower across timesteps $t$ or noise $\epsilon$. Consequently, diffusion-based classifiers evaluate multiple samples to obtain a stable estimate. Figure 4 in their paper evaluates different timestep-sampling strategies, with each curve corresponding to a distinct sampling scheme. Their analysis concludes that uniform sampling across timesteps yields the best performance.
>
> In contrast, our candidate pruning strategy leverages the next-scale prediction property of the VAR model, enabling a substantially more efficient pruning procedure that does not require a full forward pass. Unlike [f], which explores various timestep-sampling strategies, we instead focus on the strategy of using the first $K’$ scales, as this is the most computationally efficient approach. More specifically, using only the $k$-th scale incurs the same computational cost as using all first $k$ scales, since the computation of the likelihood at scale $k$ is conditioned on all previous scales. We empirically found that using all first k scales consistently outperforms using only the k-th scale. Therefore, to maximize efficiency gains, we focus on the current strategy, and its effectiveness is demonstrated in Figure 2 of our paper.
>
> As suggested by the reviewer, we additionally explored using more samples during the candidate pruning stage by combining likelihood smoothing with partial-scale pruning. The results, provided in Figure 4 in the Appendix of the revised manuscript, show that although using more samples can slightly improve top-10 pruning accuracy, it substantially increases computational cost. Therefore, we recommend using a single sample at this stage to achieve better efficiency.
>
> [f] Your Diffusion Model is Secretly a Zero-Shot Classifier, ICCV 2023
>
> > Ablation on Smoothness
>
> We would like to clarify that our observation of the unsmooth likelihood issue is based on noise sampled from a Gaussian distribution. This demonstrates that even standard Gaussian noise can cause notable perturbations in the estimated likelihood. The effectiveness of our likelihood-smoothing technique is supported by the ablation study in Table 2 of our paper, which shows a 4.94% increase in accuracy. For additional clarity, we further analyze the effect of different values of $S$, please refer to the common questions above for more details.
>
> > Unfair Comparison (Class-Incremental Learning)
>
> As our goal is to demonstrate the advantage of generative classifiers over discriminative classifiers for class-incremental learning, we follow the setup in [g] and replace only the VAE component with the VAR model. We therefore agree that generative classifiers based on other generative models could also serve as viable solutions for class-incremental learning. To verify this point, we include a diffusion classifier based on the DiT-S/2 model in our experiments. Please see the common questions above for more details.
>
> We would also like to clarify that our experimental setup is different from ControlNet, making it an unsuitable baseline for comparison. Given an image–label pair $(x, y)$, ControlNet finetunes a trainable copy of the model parameters to obtain $p_\theta(x \mid y, c)$, where $c$ is an additional conditional input (e.g., pose or depth). This design enables controllable image generation. However, in the class-incremental learning setting, the goal is instead to learn $p_\theta(x_{new} \mid y_{new})$ for samples $(x_{new}, y_{new})$ from a completely new, previously unseen class. This difference may make parameter-sharing strategies more complicated than in ControlNet, whose design focuses on adapting only the control branch.
>
> We agree that parameter-sharing techniques are crucial for reducing parameter growth when adapting to new classes, and that finding an effective design is non-trivial. This involves deciding which layers to adapt and considering alternatives such as LoRA finetuning or full-rank adapters. Since our focus in this work is to highlight the advantages of VAR-based classifiers, a deeper exploration of these architectural design choices is beyond our current scope and represents an important direction for future work.
>
> [g] Class-Incremental Learning with Generative Classifiers, ICCV 2021 workshop

---

> > ### Author Response · Authors · 2025-11-21
> >
> > > Unfair Comparison (DiT)
> >
> > We choose DiT in our experiments because it is the model adopted in the original diffusion classifier paper, which allows us to conduct a fair and direct comparison with this well-established baseline.
> >
> > Regarding the suggestion on parallelization, we increased the batch size from 25 to 2500 so that each sample can be evaluated with a single forward pass. This increases GPU memory usage from 4.407,GiB to 42.594,GiB, while reducing the estimated runtime only from 57:41:40 to 52:34:10. The limited runtime improvement is likely due to compute-bound. We have updated the runtime accordingly in the revised paper.
> >
> > We also agree that FLOPs is an appropriate metric for comparing computational efficiency, and we have incorporated it into the revised paper. The updated results in Table 1 show that our method requires 89$\times$ less computational cost compared to the DiT-based baseline. Although a single forward pass of DiT and VAR costs 114 GFLOPs and 140 GFLOPs respectively, the overall computation remains significantly lower for the VAR-based approach, demonstrating its advantage in efficiency.
> >
> > > Unfair Comparison (Rectified flow)
> >
> > To investigate whether rectified-flow models can further improve the efficiency of diffusion classifiers, we conducted a preliminary evaluation on the first 10 classes of ImageNet. We selected MeanFlow [h] as the representative rectified-flow model, since it achieves an FID of 3.45 on ImageNet-256 with a single sampling step, and compared it with the DiT-based classifier under a 5-step setup. Surprisingly, while the DiT baseline achieves an accuracy of 82.6%, the MeanFlow-based classifier attains only 40.0%.
> >
> > Our analysis indicates that the output of MeanFlow has higher prediction error compared to the DiT outputs. One possible explanation is that rectified-flow models are trained to model marginal velocity fields using supervision from conditional flows (as illustrated in Figure 2 of [h]). This mismatch may introduce a discrepancy between the instance-level conditional flow required for classification and the predicted marginal flow produced during inference. Although this training paradigm enables highly efficient sampling for generation tasks, the induced discrepancy may negatively impact classification performance.
> >
> > This observation suggests that effectively leveraging the strong sampling efficiency of rectified-flow models for classification remains an underexplored direction.
> >
> >
> > [h] Mean flows for one-step generative modeling. Arxiv 2025
> >
> > > Novel classifier
> >
> > Thank you for the suggestion. We have updated the statement accordingly.
> >
> > > Diffusion classifier is not popular
> >
> > We absolutely agree that generative classifiers, especially diffusion classifiers, are receiving increasing attention, as reflected in our references and in our motivation to explore a VAR-based approach. The only relevant statement we identified in our manuscript is: “Although generative classifiers are less commonly adopted due to the inherent difficulty of accurately modeling $p(x \mid y)$.” We would like to clarify whether this statement is what the reviewer considers a strong claim, or if there are other sentences we may have overlooked.

---

> > > ### Comment · Reviewer_1Nsc · 2025-11-24
> > > **Final comments**
> > >
> > > I would like to keep my score the same for the following reasons
> > >
> > > This paper is a soup of ideas where VAR model works. There is no connection between explainability and generative models,. A better title would have been YOUR VAR MODEL IS AN EXPLAINABLE MODEL AND YOUR VAR MODEL IS A GENERATIVE CLASSIFIER.
> > >
> > > The novelty of the work is limited as also pointed by other reviewers.
> > >
> > > Authors try different problems with VAR but do not explore them deeply. For example, at the task of class incremental learning, there are many generative classifiers / any method that excel at this task but no comparison or in what areas or settings specifically VAR classifier outperforms. In addition, From Table 4, DC performs better than A-VAR+. There is no explanation provided for this result.
> > >
> > > Efficiency is a property of VAR models, not sure this can be claimed as novelty of the generative models.
> > >
> > > Visual explainability is a promising area specifically to VAR, thanks to its exact like-hood. However, there is no connection to the classifier. Analysis on what are the advantages of VAR explainability compared to other explainability such as attention maps, approximate like-hood of the diffusion models. more insights on this would improve the paper.
> > >
> > > In addition, when making changes to the manuscripts try to use a different color to indicate the change.

---

> > > ### Comment · Reviewer_1Nsc · 2025-11-24
> > > **Clarifications**
> > >
> > > 1. Experiments directly on rectified flow would have been more insightful.
> > > 2. Large chunk of time would be spent on moving data to the GPUs, I disagree with the authors comments on compute intensive. I suspect the diffusion model is not well tuned. Authors code would have been more helpful. Large enough GPUs with parallellization gives you results in one batch pass, equivalent to VAR.

---

> ### Author Response · Authors · 2025-11-27
>
> We would like to thank the reviewer for the constructive feedback and the engagement in the discussion. We provide our responses as follows.
>
> > Different color for changes
>
> We have highlighted the meaningful changes from the initial submission using different colors, excluding minor wording or formatting edits.
>
> > Connection between classifier and explanation
>
> We agree that the VAR model itself provides inherent explainability. However, we would like to clarify a different perspective regarding the claim that ‘there is no connection to the classifier.’ In our case, the VAR model performs both classification and explanation through the same likelihood function, establishing a strong and direct connection between the two. This close coupling is not present in post-hoc explanation methods such as Grad-CAM, where the explanation relies on gradients that do not directly correspond to the classifier’s final decision.
>
> Therefore, rather than having ‘no connection to the classifier,’ the VAR-based generative classifier actually exhibits a tighter alignment between prediction and explanation compared to traditional post-hoc methods.
>
> > Lack of novelty
>
> We would like to reiterate the main contributions and novelty of our work.
>
> **(1) First exploration of VAR-based generative classifiers.**
>
> To the best of our knowledge, this is the first work to investigate a VAR-based generative classifier, demonstrating its strong efficiency and unique properties. We believe this opens a new, previously under-explored direction—similar to how [f] introduced diffusion classifiers and subsequently inspired numerous follow-up works. This originality is also recognized by other reviewers.
>
> [f] Your Diffusion Model is Secretly a Zero-Shot Classifier, ICCV 2023
>
> **(2) A-VARC$^+$: Substantial performance and efficiency gains.**
>
> We show that a naïve VAR-based classifier using Bayes’ inference is suboptimal. Building on our empirical observations, we introduce likelihood smoothing, partial-scale candidate pruning, and CCA-based enhancement to form A-VARC$^+$. On ImageNet-100, these techniques improve accuracy by 6.02% while reducing computational cost by 3×. We believe such improvements go well beyond a trivial enhancement.
>
> **(3) Comprehensive and well-controlled comparison across generative model families.**
>
> We thoroughly analyze normalizing flow-, diffusion-, VAR-, and (now added) rectified flow-based generative classifiers across multiple datasets, offering insights into their respective strengths and limitations.
>
> **(4) First demonstration of tractable likelihood for visual explanation.**
>
> We are the first to show that the tractable likelihood in VAR enables token-wise attribution for visual explainability, with both qualitative and quantitative evaluations demonstrating promising interpretability results.
>
> **(5) Furthering the study of generative classifiers for class-incremental learning.**
>
> While prior work has explored the concept using VAE, we extend the investigation using state-of-the-art generative models (VAR and diffusion) and provide analysis on their usefulness in class-incremental learning scenarios.
>
> Given these contributions, we believe the critique regarding lack of novelty may be overly strong. Our work not only advances the performance and efficiency of generative classifiers but also provides new insights that can benefit future research in this direction.
>
> > Class-incremental learning setup
>
> Catastrophic forgetting is a central challenge in class-incremental learning, and therefore most existing methods assume access to a replay buffer. In our experiments, however, we adopt a more challenging setup where no replay data is available. This allows us to clearly demonstrate the advantage of generative classifiers over conventional discriminative approaches in such a realistic and difficult scenario.
>
> As shown in Table 4, the proposed A-VARC$^+$ achieves the best average accuracy, which is the primary evaluation metric for class-incremental learning. We emphasize that our objective is not to argue that a VAR-based approach is universally superior; rather, our goal is to show that class-incremental learning challenges can be effectively addressed using modern generative models. This highlights their strong potential in continual-learning applications.
>
> > Experiment of rectified flow
>
> As suggested, we have included the results and discussion of rectified flow-based methods in the updated manuscript. Our experiments show that, under the same number of samples, the rectified flow-based classifier consistently performs worse than the DiT counterpart across all evaluated datasets. Please refer to the updated Table 1 for the detailed results.

---

> > ### Author Response · Authors · 2025-11-27
> >
> > > GPU runtime
> >
> > Thank you for the clarification. The diminishing efficiency gains when increasing batch size are also reported in other works, and [i] discusses how to identify an optimal batch size for maximizing throughput. This observation is consistent with our findings that simply increasing batch size results in only limited efficiency improvement. Achieving further gains would require more advanced strategies such as model replication, which introduces implementation complexity.
> >
> > To avoid ambiguity arising from implementation details, and following the reviewer’s suggestion, we now report only GFLOPs as the computational cost metric and have removed runtime comparisons. GFLOPs directly quantify the core computation required by each method and thus provide a fairer and implementation-agnostic basis for comparison.
> >
> > We hope this fully resolves the concerns regarding implementation-dependent runtime evaluation.
> >
> > [i] Mind the Memory Gap: Unveiling GPU Bottlenecks in Large-Batch LLM Inference, IEEE CLOUD 2025.

---

### Official Review · Reviewer_kzRp · 2025-10-28

**Soundness:** 3
**Presentation:** 2
**Contribution:** 2
**Rating:** 4
**Confidence:** 4

**Summary:**

The paper proposes a VAR-based generative classifier (A-VARC+) that adds likelihood smoothing, partial-scale candidate pruning, and CCA finetuning. It argues for better efficiency than diffusion-based generative classifiers and offers token-level PMI explanations and a small class-incremental learning demo.

**Strengths:**

1. Clear motivation for exploring autoregressive generative classifiers.

2. Techniques are simple, practical, and likely easy to reproduce.

3. Tractable likelihood enables straightforward token-wise visual explanations.

4. Broad empirical sweep (in-distribution + several shift datasets).

**Weaknesses:**

1. Technical novelty feels incremental; mainly a combination/adaptation of known ideas.

2. Accuracy advantages are limited; strong discriminative baselines still perform better.

3. Robustness/CL claims are weakly evidenced (small-scale setups, mostly qualitative).

4. Limited analysis of trade-offs (e.g., compute/memory vs. accuracy) and few quantitative explainability metrics.

**Questions:**

see weakness

---

> ### Author Response · Authors · 2025-11-21
>
> We thank the reviewer for the insightful review and constructive comments. We provide our responses as follows.
>
> > Technical novelty feels incremental.
>
> We would like to emphasize that our goal is to advance the research direction of generative classifiers. Since this area remains relatively underexplored, prior works such as [d, e] primarily focus on understanding the properties and potential of generative classifiers rather than proposing highly sophisticated technical innovations, and these efforts have still been well recognized by the community. In the same spirit, our work investigates the VAR-based generative classifier and demonstrates its substantial efficiency advantages over diffusion-based approaches, thereby improving the practical applicability of generative classifiers. Furthermore, we highlight additional benefits of VAR-based classifiers, including visual explainability and natural compatibility with class-incremental learning due to their tractable likelihood. We believe these findings provide meaningful insights and contribute to the continued development of generative classifiers.
>
> [d] Intriguing properties of generative classifiers, ICLR 2024
>
> [e] Generative Classifiers Avoid Shortcut Solutions, ICLR 2025
>
> > Accuracy advantages are limited
>
> We acknowledge that, in terms of raw classification accuracy, current generative classifiers—including both diffusion-based and VAR-based methods—are still inferior to well-established discriminative classifiers. However, generative classifiers offer unique advantages such as inherent visual explainability and the ability to adapt naturally to class-incremental learning.
>
> > Robustness/CL claims are weakly evidenced
>
> We provide quantitative results for class-incremental learning in Table 4 of our paper. We agree that the experimental setup is relatively small, as it is intended to serve as a proof of concept demonstrating that a VAR-based classifier can naturally support class-incremental learning. Despite the limited scale, the results clearly highlight a key advantage of VAR-based generative classifiers: they can adapt to new classes without requiring additional replay data, underscoring their suitability for class-incremental learning scenarios.
>
> To further examine whether this property extends to other generative classifiers, we included a diffusion classifier based on the DiT-S/2 model in our updated experiments during the rebuttal. Please refer to the common questions above for the results.
>
> > Limited analysis of trade-offs and few quantitative explainability metrics.
>
> Please refer to the common questions above for the quantitative analysis of visual explainability.
>
> Regarding the trade-off analysis, Figure 2 in our paper illustrates a clear relationship between accuracy and computational cost. We also provide additional analysis such as
> likelihood smoothing with varying value of S and $\sigma$ as well as combination of candidate pruning and likelihood smoothing in Sec. B of Appendix of the updated manuscript. We hope these results provide sufficient guidance for users to adaptively choose the trade-off that best suits their application.

---

### Official Review · Reviewer_XVX3 · 2025-10-31

**Soundness:** 3
**Presentation:** 3
**Contribution:** 3
**Rating:** 8
**Confidence:** 5

**Summary:**

This paper considers the problem of generative classification with an autoregressive model. Traditional discriminative classifiers directly model $p(y|x)$, while a generative model can be turned into a classifier via $\text{argmax} p(x|y)$. This paper proposes a pipeline that has 3 steps: 1) likelihood smoothing (add gaussian noise to the latent features), 2) partial-scale pruning (use first few scales to give rough likelihood estimates for a few classes), and 3) CCA  coupled with a multi-scale autoregressive image model to perform classification. More specifically, to turn an autoregressive image model into a classifier, for each class, we then compute the class conditional log likelihood using the already trained model, and then pick the class with the highest likelihood. The paper shows that this pipeline is resistant to catastrophic forgetting, and that brings a bonus of interpretability.

**Strengths:**

The key inspiration behind this paper is the need to move away from a diffusion-centric approach that has limited efficiency. Here are concrete strengths of this work:

- The paper evaluates comprehensively on the ImageNet dataset testing ImageNetV2, R, etc. More concretely, the method proposed here actually is close in top-1 accuracy to baseline methods but almost 2 orders of magnitude faster.
- The pipeline proposed here is quite simple and easy to put together for common autoregressive models. The insight that using an autoregressive model allows for a factorization of the exact likelihood that is a nice one, and the use of the next-scale prediction approach from Tian et. al. is critical for this work.
- The interpretability benefits from this work also are impressive. Because VAR models have tractable likelihoods, the paper define token-wise pointwise mutual information (PMI) to quantify how much each token (or image patch) contributes to the predicted class. This is an interesting side benefit of this approach.
- Finally, the paper is quite well written. In particular, the introduction and the related work guide the reader through the motivation for this work.

**Weaknesses:**

The weaknesses are provided below:
- The focus of the evaluation is primarily on the Imagenet dataset using 50 images per class. Is this standard in this literature? I wonder whether more extensive evaluations can be performed in other settings.
- The interpretability claims are interesting, but would need more than qualitative images to thoroughly validate.
- CCA Fine-tuning might be doing discriminative training in disguise. See the question section for more details on this.

**Questions:**

What is the discriminative finetuning method doing? To be more concrete, I think we can interpret this loss function (equation 11) as a contrastive discriminative objective. If we define the score: $s_\theta(x, y) = \log p_\theta(x|y)$, then under uniform priors, the posterior can be expressed as a softmax over scores:
$$ p_\theta(y|x) = \frac{e^{s_\theta(x, y)} }{\sum_y's_\theta(x, y')}.$$
We can then say that the CCA enforces: $-\log  \sigma( \beta[ s_{\theta(x, y)} - s_{\theta(x, y_{\text{neg}})} ])$. This is equivalent to the binary logistic loss used in discriminative contrastive learning setting. To clarify in words, it might be the case that: CCA is discriminative training on log-likelihood ratios, using the generator’s likelihood as the scoring function. This means: CCA finetuning is functionally identical to discriminative contrastive learning, but applied to generative likelihoods rather than logits.

Questions:
- Wouldn't this bias the samples from the generative models towards class discrimination rather than sample realism?

---

> ### Author Response · Authors · 2025-11-21
>
> We thank the reviewer for the insightful review and constructive comments. We provide our responses as follows.
>
> > Is evaluation on 50 images per class a standard setup?
>
> The ImageNet validation set contains 50,000 images, consisting of 1,000 classes with 50 samples per class; thus, using 50 images per class is the standard evaluation protocol. In our experiments, we evaluate on a subset of 100 classes to enable a fair comparison with approaches that have significantly higher computational cost.
>
> > The interpretability needs further validation
>
> We have added additional qualitative results in the revised manuscript. We also provide quantitative analysis during the rebuttal phase; please refer to the common questions above for details.
>
> > CCA Fine-tuning might be doing discriminative training in disguise.
>
>  It is a very good question. Although the spirit of the CCA loss is related to discriminative contrastive loss, their behaviors differ substantially due to the involvement of a fixed pretrained model. In our experiments, we found that directly training with a discriminative contrastive loss leads to catastrophic degradation of the model’s generative ability. This occurs because the discriminative contrastive loss enforces only relative differences between $s_\theta(x, y)$ and $s_\theta(x, y^{neg})$, without preserving the absolute scale of the conditional likelihood.
>
> In contrast, the CCA objective incorporates an additional fixed pretrained model, which constrains the conditional likelihood of positive pairs from collapsing. Omitting hyperparameters for simplicity, the CCA loss can be rewritten as
>
> $-\log\sigma(s_\theta(x,y) - s_\phi(x,y))-\log\sigma(s_\phi(x,y^{neg}) - s_\theta(x,y^{neg}))$
>
> from which we can see that the log-likelihood is effectively optimized relative to a fixed pretrained model $s_\phi()$. This design ensures that the conditional likelihood of the positive pair does not degrade too much from the pretrained model, thereby preserving the generative quality of the VAR model. As reported in the CCA paper, the FID scores actually improve after finetuning, demonstrating that the CCA objective can simultaneously benefit both generation and classification.

---

### Official Review · Reviewer_d2kX · 2025-11-01

**Soundness:** 3
**Presentation:** 3
**Contribution:** 2
**Rating:** 6
**Confidence:** 4

**Summary:**

Given the existing diffusion classifiers, Adaptive VAR Classifier$^+$ is proposed in this paper as a new variant of generative classifiers. It is claimed to be  efficient, visually explainable and robust to distribution shifts. Experiments are conducted accordingly.

**Strengths:**

1. **Clarity**: This work is clearly presented with the linear logic. The core motivation and contributions of Adaptive VAR Classifier$^+$ is articulated with examples and comprehensive comparisons to existing generative classifiers. The mechanism of the model is also coherently explained.

2. **Originality**: It is not previously proposed and studied on VAR classifiers. In past research, VAR and its variants are workhorses for generative tasks. To the best of my knowledge, this work is from the original attempts and findings of VAR as classifiers.

**Weaknesses:**

1. The core novelty of this work should be improved. The idea to propose VAR as a generative classifier follow the same logic as substituting a module in an existing model with a more recent one, in specific, replacing diffusion models in diffusion classifiers with VAR. In Abstract, it should be articulated clearly on what the pressing problem is in classification for this era of large generative models and why proposing VAR as a generative classifier can solve this problem to highlight the significance.

2. From results in Table 1, it seems that for the robustness against distribution shifts, Adaptive VAR Classifier$^+$ does not demonstrate clear advantage compared to diffusion classifiers and traditional classifiers such as ResNets. The claimed contributions on robustness to distribution shifts are not sufficiently achieved in this case.

**Questions:**

1. For likelihood smoothing, why are Gaussian noises used to avoid the sensitivity to the small perturbations introduced by adding Gaussian noises? Could the authors provide an intuitive explanation? Compared to $\epsilon$ in Equation (8), is $\epsilon_i$ in Equation (9) scaled? In numerical experiments, could the authors provide an ablation study on $S$?

2. Why is the novelty to apply CCA in the proposed Adaptive VAR Classifier$^+$? It seems that CCA is already proposed and published in Toward Guidance-Free AR Visual Generation via Condition Contrastive Alignment [1] which is also for autoregressive models.

[1] Huayu Chen, Hang Su, Peize Sun, and Jun Zhu. Toward guidance-free ar visual generation via condition contrastive alignment. arXiv preprint arXiv:2410.09347, 2024b.

---

> ### Author Response · Authors · 2025-11-21
>
> We thank the reviewer for the insightful review and constructive comments. We provide our responses as follows.
>
> > The core novelty of this work should be improved.
>
> Thank you for the suggestion. We have updated the paper accordingly, focusing on the efficiency gain of the VAR-based approach and the benefits brought by tractable likelihood.
>
> > The claimed contributions on robustness to distribution shifts are not sufficiently achieved in this case.
>
> We would like to clarify that, unlike diffusion-based methods, the VAR-based approach does not exhibit a clear advantage in robustness to distribution shift. However, we show that the VAR-based generative classifier offers visual explainability and serves as a promising solution for class-incremental learning.
>
> > Intuition of likelihood smoothing.
>
> The intuition behind the smoothing technique is as follows. By adding a small Gaussian noise to the latent, we obtain a perturbed latent that remains close to the original one. Repeating this process several times yields multiple neighboring latents. To encourage smoothness in the estimated likelihood, we replace the likelihood of the original latent with the average likelihood across these neighboring latents, effectively smoothing the estimate within the local neighborhood.
>
> The size of this neighborhood is controlled by $\sigma$. We provide an ablation study with different $\sigma$ values: a small $\sigma$ results in only minimal smoothing, while a large $\sigma$ risks averaging over semantically different samples. Empirically, we find that $\sigma = 0.1$ offers a good balance.
>
> For the ablation study on different values of $S$, please refer to the common-response section above.
>
> | Sigma | 0.01 | 0.05 | 0.1 | 0.5 | 1.0 |
> |-------|------|------|-----|-----|-----|
> | Acc   | 85.50 | 87.72 | 88.26 | 84.38 | 39.54 |
>
> > What is the novelty of applying CCA in the proposed Adaptive VAR Classifier?
>
> Our contribution is to show that the CCA objective, originally proposed to improve class-conditional image generation, can also benefit classification performance. We believe this observation is non-trivial, as another widely used technique, classifier-free guidance, similarly improves class-conditional generation but can degrade classification accuracy. Our results therefore highlight that it is possible for a single objective to enhance both generation and classification.
>
> We acknowledge that the CCA technique itself originates from prior work and should not be considered our main contribution. Accordingly, we clearly cite the original source and present it as an add-on component that further boosts performance.

---

### Author Response · Authors · 2025-11-21
**Common Questions**

We would like to thank all the reviewers for their constructive and insightful comments. Below, we address the common questions raised by multiple reviewers in a unified manner.

> Quantitative analysis of the visual explainability.

In this work, we present using token-wise mutual information (TMI) for visual explanation. This can be viewed as an attribution method that produces an attribution score for each token. To evaluate attribution quality, we adopt the insertion and deletion metrics introduced in [a], which are widely used in the explainability literature. Tokens are first sorted according to their attribution scores; then, they are gradually inserted or removed to measure the change in the predicted probability of the ground-truth class. The area under the curve (AUC) is used for evaluation. Intuitively, for the insertion metric, a higher AUC is preferred, as it indicates that tokens with high attribution scores meaningfully support the ground-truth class. For the deletion metric, a lower AUC is preferred, as removing highly attributed tokens should decrease the ground-truth probability. We additionally report LIME [b] and SHAP [c] as baselines. Please refer to Sec. C in the Appendix of the revised paper for additional details.

For A-VARC, LIME achieves the best overall performance, while TMI performs comparably to SHAP on the insertion metric and achieves the second-best result on the deletion metric. For A-VARC$^+$, TMI achieves the best performance on both insertion and deletion. This result aligns with the qualitative observation in Figure 5 of our paper: after finetuning, TMI becomes more concentrated on class-relevant regions.

* Result of A-VARC
| Method      | LIME      | SHAP     | TMI (Ours) |
|-------------|-----------|----------|------------|
| **Insertion (↑)** | **0.979** | 0.853 | 0.845   |
| **Deletion (↓)**  | **0.192** | 0.432 | 0.346   |

* Result of A-VARC$^+$
| Method      | LIME      | SHAP     | TMI (Ours) |
|-------------|-----------|----------|------------|
| **Insertion (↑)** | 0.939 | 0.902 | **0.944** |
| **Deletion (↓)**  | 0.614 | 0.746 | **0.605** |

[a] RISE: Randomized Input Sampling for Explanation of Black-box Models, BMVC 2018

[b] "Why Should I Trust You?": Explaining the Predictions of Any Classifier, KDD 2016

[c] A Unified Approach to Interpreting Model Predictions, NeurIPS 2017

> Ablation study of different S for likelihood smoothing

Following the setting in Table 2, we provide an additional ablation study using different values of $S$. For A-VARC, the accuracy increases as $S$ grows and saturates at $S = 16$, yielding an overall improvement of 5.12%. For A-VARC+, saturation occurs earlier at $S = 8$ with a smaller improvement of 1.06%, indicating that the gain from smoothing is reduced when combined with CCA. In both cases, the results consistently show that applying likelihood smoothing improves classification accuracy.

| Method    | S=1   | S=2   | S=4   | S=8   | S=10  | S=16  | S=32  |
|-----------|-------|-------|-------|-------|-------|-------|-------|
| A-VARC    | 83.30 | 85.30 | 87.18 | 88.18 | 88.26 | 88.42 | 88.42 |
| A-VARC+   | 88.68 | 89.42 | 89.30 | 89.74 | 89.72 | 89.54 | 89.72 |

> Diffusion classifier for class-incremental learning

To investigate whether other generative classifiers can address the class-incremental learning task, we include a diffusion classifier based on DiT-S/2 (we observed that larger models tend to overfit in this small-scale setting) and train it for 2,000 epochs. The results show that the diffusion classifier can also adapt naturally to this setting and achieves performance comparable to the VAR-based method.
| Method | None | CWR | DC | A-VARC+ |
|--------|------|------|-------------|--------|
| Task1  | 0.0  | 83.2 | 78.4 | 72.4 |
| Task2  | 82.4 | 61.6 | 73.6 | 82.4 |
| Avg    | 41.2 | 72.4 | 76.0 | **77.4** |

---

### Meta-Review · Area_Chair_xESr · 2026-01-07

**Summary:**

Reviewers broadly agree that the paper is clearly written, technically sound, and explores an interesting direction. However, they raise recurring concerns regarding (i) the incremental nature of the technical novelty, (ii) overstated or weakly supported claims on the proposed generative classifier, (iii) whether efficiency and explainability stem from VAR models in general rather than from the proposed method, and (iv) the depth and fairness of empirical validation.

The rebuttal adds several ablations, clarifies claims, and corrects earlier overstatements, which improves the empirical completeness and correctness of the work. These changes primarily strengthen the support for existing claims, and address the main concerns raised by the reviewers and clarify the scope and contribution of the proposed method, despite the remaining limitations in novelty.

**Reviewer Concerns:**

### Concerns addressed by the rebuttal
- **Likelihood smoothing and ablation on hyperparameters:**
Reviewers questioned the intuition behind Gaussian likelihood smoothing and the lack of ablations on smoothing strength. The authors provided clear local-smoothness interpretation, added ablations, and showed that performance improves within a reasonable range. These additions address the reviewers’ technical questions and strengthen the empirical justification for this component.
- **Quantitative evaluation of visual explainability:**
Initial explainability claims were largely qualitative. The authors added quantitative analysis and comparisons against LIME and SHAP. This substantially improves the credibility of the explainability claims.
- **Overstated claims and wording:**
The authors acknowledged these issues and updated the wording accordingly. While this does not strengthen the contribution, it improves correctness and tone and removes clear points of contention.
- **Efficiency comparison and compute metrics:**
Concerns about runtime comparisons and fairness were addressed by replacing wall-clock timing with GFLOPs and by including additional analyses on parallelization limits.

### Concerns only partially addressed or still outstanding
- **Strength and nature of technical novelty:**
Multiple reviewers characterize the work as a soup of ideas or an incremental adaptation of existing components. The rebuttal reframes novelty as the first exploration of VAR-based generative classifiers and emphasizes efficiency and tractable likelihood. While this positioning is reasonable, it does not fully counter the concern that most algorithmic elements are adaptations rather than fundamentally new techniques.
- **Robustness and distribution shift claims:**
Reviewers noted that the proposed method does not clearly outperform diffusion classifiers and traditional classifiers under distribution shift. The authors concede this point and emphasize explainability and class-incremental learning instead. This is an appropriate correction, but it also confirms that the original robustness claim was not substantiated.

**Reviewer Scores:**

**Reviewer XVX3 (Score: 8, accept):**

Strongly positive on efficiency, simplicity, and interpretability. Rebuttal additions likely reinforce this assessment.

**Reviewer d2kX (Score: 6, marginally above acceptance):**

Core novelty concerns remain, but added ablations and clarifications likely keep the score in the borderline-positive range.

**Reviewer kzRp (Score: 4, marginally below acceptance):**

Incremental novelty and limited gains remain unresolved. Rebuttal improves clarity but does not fundamentally change the assessment.

**Reviewer 1Nsc (Score: 4, marginally below acceptance):**

Explicitly maintains score after rebuttal, citing limited novelty, limited exploration of multiple claims, and lack of strong comparative evidence.

---

### Decision · Program_Chairs · 2026-01-26

Accept (Poster)